# Asymptotically exact variational flows
# via involutive MCMC kernels

**Zuheng Xu**     **Trevor Campbell**
Department of Statistics
University of British Columbia
[zuheng.xu | trevor]@stat.ubc.ca

## Abstract

Most expressive variational families—such as normalizing flows—lack practical convergence guarantees, as their theoretical assurances typically hold only at the intractable global optimum. In this work, we present a general recipe for constructing tuning-free, asymptotically exact variational flows on *arbitrary* state spaces from involutive MCMC kernels. The core methodological component is a novel representation of general involutive MCMC kernels as invertible, measure-preserving *iterated random function* systems, which act as the flow maps of our variational flows. This leads to three new variational families with provable total variation convergence. Our framework resolves key practical limitations of existing variational families with similar guarantees (e.g., MixFlows), while requiring substantially weaker theoretical assumptions. Finally, we demonstrate the competitive performance of our flows across tasks including posterior approximation, Monte Carlo estimates, and normalization constant estimation, outperforming or matching No-U-Turn sampler (NUTS) and black-box normalizing flows.

## 1 Introduction

Variational inference (VI) [1–3] is a general methodology for approximate probabilistic inference, where the goal is to approximate a target distribution (e.g., a Bayesian posterior) within a specified variational family. This variational family is typically chosen to be a parametric family that enables tractable inference—allowing for i.i.d. sampling and density evaluation [2–7]. This tractability offers key benefits: it enables the evaluation and optimization of approximation quality via unbiased estimates of the evidence lower bound (ELBO) [3], which corresponds to the Kullback-Leibler (KL) divergence [8] to the target distribution up to a constant. Moreover, it facilitates downstream tasks such as importance sampling [9, 10] and normalization constant estimation.

The quality of a variational approximation is fundamentally determined by the expressiveness of its variational family. Significant progress has been made in constructing flexible families, including boosted mixtures [11–16] and normalizing flows [4, 6, 17–20]. These families often exhibit *universal approximation guarantees* [16, 21, 22]: as the number of mixture components or flow layers grows, the family can approximate any distribution arbitrarily well under mild assumptions. However, a major limitation remains—theoretical guarantees pertain only to the *optimal* variational approximation, which is rarely obtained in practice due to non-convex optimization. In contrast, Markov chain Monte Carlo (MCMC) [23, 24; 25, Ch. 11, 12] is *asymptotically exact*, meaning it is guaranteed to produce arbitrarily accurate results given sufficient computation for any valid choice of tuning parameters (though some values may yield higher efficiency than others).

This has motivated extensive work aimed at bridging VI and MCMC. In particular, many recent approaches fall under the categories of differentiable annealed importance sampling (DAIS) or differentiable sequential Monte Carlo (DSMC) [26–36]. These methods can be interpreted as gradient-based tuning of AIS/SMC exploration or backward kernels to improve approximation quality. How-

ever, their theoretical guarantees in the limit of flow length [29, 31, 37] often rely on idealized assumptions—such as perfect transitions or diminishing stepsizes—that rarely hold in practice or only apply under optimal tuning. Moreover, AIS/SMC methods are known to be highly sensitive to tuning, and DAIS/DSMC methods inherit substantial tuning cost to ensure robust performance [38].

Xu et al. [39] introduced MixFlow, an asymptotically exact variational family that does not require optimal tuning. A MixFlow is constructed by averaging pushforwards of a reference distribution under repeated application of an invertible map. When this map is both *ergodic* and *measure-preserving* (e.m.p) with respect to the target distribution $\pi$, MixFlows converge to $\pi$ in total variation as the number of steps increases, while retaining the tractability of standard variational inference. However, its practical applicability is limited by the challenge of designing an invertible $\pi$-e.m.p. map for general continuous targets (several solutions exist for discrete spaces [40, 41]). The main obstacles are: (1) continuous e.m.p. maps often involve simulation of ODEs, which requires discretized numerical methods that destroy the e.m.p. property; (2) exactness often requires discrete Metropolis–Hastings (MH) corrections that are not invertible; and (3) proving ergodicity of such maps is very challenging. For example, Xu et al. [39] proposed a map based on the *uncorrected* Hamiltonian Monte Carlo (HMC), which is neither exactly measure-preserving nor provably ergodic. Other existing Hamiltonian-based methods [42] also suffer from discretization error and are non-ergodic [43]. Attempts via deterministic Gibbs samplers based on measure-preserving ODEs [44] or CDF/inverse-CDF transformations [40] are also limited by the intractability of computing the exact transformations. MH corrections used to restore exactness [40, 45] result in non-invertible transformations due to the accept-reject mechanism; recall that invertibility is required by variational flows to enable tractable density evaluation. To date, there is no framework for constructing variational families whose *practical implementation* achieves an MCMC-like asymptotic exactness.

In this work, we address the challenges mentioned above and propose a new framework for developing practical, asymptotically exact variational flows. Rather than relying on e.m.p dynamics as in MixFlow [39], our framework leverages *iterated random functions* (IRF)[1] [47]—a type of *random dynamical system*. The main contributions of this work are as follows:

1. We develop a method for deriving exact measure-preserving transformations from general *involutive MCMC* kernels [48, 49], while preserving invertibility of the transformation.

2. We introduce a more general framework for constructing asymptotically exact flows, leading to three novel variational families beyond the original MixFlow for general state spaces.

3. We establish total variation convergence guarantees for these new families under significantly weaker assumptions than those required in MixFlow theory [39], notably relaxing the ergodicity conditions of the flow maps.

## 2 Background

Throughout, let $\pi$ be a target distribution on a measurable space $(\mathcal{X}, \mathcal{B})$ equipped with a $\sigma$-finite base measure $m$. All distributions are assumed to have densities with respect to the base measure on their corresponding spaces, and we use the same symbol to denote both a distribution and its density. Given a transformation $f$ and a distribution $p$, we write $f(p(x))$ for the function $f$ evaluated at $p(x)$, and $fp(x)$ for the density of the pushforward distribution $fp$ evaluated at $x$.

### 2.1 Homogeneous MixFlows

A mixed variational flow (MixFlow) [39, 41] is built from a *deterministic*, $\pi$-ergodic (Definition D.1) and measure-preserving (e.m.p.) diffeomorphism $f$[2]. Given such a map $f$ and a reference distribution $q_0$ on $\mathcal{X}$ that enables i.i.d. sampling and density evaluation, the MixFlow density is given by

$$\forall x \in \mathcal{X}, \quad \bar{q}_T(x) = \frac{1}{T} \sum_{t=1}^{T} f^t q_0(x) = \frac{1}{T} \sum_{t=1}^{T} \frac{q_0\left(f^{-t}x\right)}{\prod_{i=1}^{t} J\left(f^{-i}x\right)}, \quad J(x) = |\det \nabla f(x)|, \quad (1)$$

where $f^t x$ and $f^t q_0$ denote mapping $x$ or pushing $q_0$ through $t$ ($t > 0$) iterations of $f$. We use the convention that $\bar{q}_0 = q_0$ (MixFlow of length 0 is just the reference distribution $q_0$). Eq. (1) is tractable if $f^{-1}$ and the Jacobian $J$ can be evaluated. To generate $X \sim \bar{q}_T$, we first draw $X_0 \sim$

---

[1]IRFs are also referred to as iterated function systems (IFS) in some literature, e.g., [46].

[2]$f$ is a diffeomorphism if it is continuously differentiable and has a continuously differentiable inverse.

$q_0$ and a flow length $K \sim \mathrm{Unif}\{1, 2, \ldots, T\}$, and then map $X_0$ through $K$ iterations of $f$, i.e., $X = f^K(X_0)$. Since $\bar{q}_T$ is built from a time-homogeneous e.m.p dynamical system, we label it a *homogeneous MixFlow*, to distinguish it from our proposed *random* dynamical system flows (see Section 4). The asymptotic exactness of homogeneous MixFlows comes from the fact that $\lim_{T\to\infty} \mathrm{TV}(\bar{q}_T, \pi) = 0$ regardless of the tuning of the flow map $f$ [39, Theorem 4.2].

In practice, the map $f$ is typically designed to mimic familiar MCMC kernels [39, 41], so that its trajectories have similar statistical behavior to the corresponding Markov chain. Despite this, general constructions of *exact* e.m.p. MixFlow maps for continuous target distributions remain unavailable. As discussed in the introduction, achieving both exact measure preservation and ergodicity is highly non-trivial in practice. Consequently, practitioners often rely on approximate maps, leading to a gap between theoretical guarantees and practical implementations. These approximations can introduce numerical instability and degrade performance as $T$ increases [39, 50]. In Section 4.1, we show how to design homogeneous MixFlows that are exact in practice. Additionally, we present a refined characterization of the density $\bar{q}_T$ by leveraging the measure-preserving property of $f$, which simplifies implementation, improves robustness, and provides a more intuitive convergence analysis.

## 2.2 Involutive MCMC

An involutive MCMC kernel [48, 49, 51] is a Metroplis-type Markov kernel with a deterministic proposal defined by an *involution* $g$, i.e., a self-inverse function satisfying $g = g^{-1}$. This framework encompasses a broad class of MCMC algorithms, with many popular algorithms appearing as special cases [49, 52–55] (see Appendix A.1). The detailed transition procedure of involutive MCMC is described in Algorithm 1 of Appendix A.2. Consider an auxiliary variable $v$ defined on a space $\mathcal{V}$, with conditional density $\rho(v \mid x)$ given $x \in \mathcal{X}$ with respect to a base measure $m_v$ on $\mathcal{V}$, and the augmented target density $\bar{\pi}(x, v) := \pi(x)\rho(v|x)$. Let $\overline{m} := m \times m_v$ be the joint base measure on $\mathcal{X} \times \mathcal{V}$. For an involution $g : \mathcal{X} \times \mathcal{V} \to \mathcal{X} \times \mathcal{V}$, each transition from state $x$ proceeds in three steps:

1. Sample an auxiliary variable $v \sim \rho(\mathrm{d}v \mid x)$;
2. Propose a new state $(x', v') = g(x, v)$;
3. Accept $x'$ with probability $\min\left(1, \frac{\bar{\pi}(x', v')}{\bar{\pi}(x, v)} J_g(x, v)\right)$ where $J_g(x, v) := \frac{\mathrm{d}g\overline{m}}{\mathrm{d}\overline{m}}(x, v)$[3].

An involutive Markov kernel $K$ defined this way is *reversible* with respect to both the augmented target $\bar{\pi}(x, v)$ and its marginal $\pi(x)$ [51, Theorem 2].

**Proposition 2.1.** *The involutive MCMC kernel $K(x', v'|x, v)$ (defined in Algorithm 1) satisfies that*

$$K(x', v'|x, v)\bar{\pi}(x, v) = K(x, v|x', v')\bar{\pi}(x', v'), \quad \widehat{K}(x'|x)\pi(x) = \widehat{K}(x|x')\pi(x'),$$

*where $\widehat{K}$ is the marginalized kernel defined as: $\widehat{K}(x'|x) := \int \widehat{K}(x', v' \mid x, v)\rho(\mathrm{d}v|x)\mathrm{d}v'$.*

## 2.3 Iterated random functions

An *iterated random function* (IRF) system [47] on $\mathcal{X}$ consists of a sequence of *random* maps:

$$\forall t \in \mathbb{N}, \ X_{t+1} = f_{\theta_{t+1}}(X_t), \quad X_0 \in \mathcal{X}, \quad (\theta_t)_{t\in\mathbb{N}} \overset{\mathrm{iid}}{\sim} \mu, \tag{2}$$

where $\{f_\theta : \mathcal{X} \to \mathcal{X} : \theta \in \Theta\}$ is a set of parametrized functions, with each $\theta$ drawn *randomly* from a distribution $\mu$ on the parameter space $\Theta$. The above IRF induces a Markov kernel given by:

$$\forall x \in \mathcal{X}, \quad \forall B \in \mathcal{B}, \quad P(x, B) := \int_\Theta \mathbb{1}_B(f_\theta(x))\mu(\mathrm{d}\theta). \tag{3}$$

This yields a simple characterization of the action of the Markov process $P$ on a distribution $q$:

$$Pq(y) := \int_\mathcal{X} P(x, y)q(\mathrm{d}x) = \mathbb{E}\left[f_\theta q(y)\right], \quad \theta \sim \mu, \quad f_\theta q: \text{pushforward of } q \text{ under } f_\theta. \tag{4}$$

Throughout this work, we focus on IRFs where the family $\{f_\theta : \theta \in \Theta\}$ satisfies Assumption 2.2.

**Assumption 2.2.** *For $\mu$-a.s. all $\theta \in \Theta$, $f_\theta(\cdot)$ is bijective and $\pi$-measure-preserving ($\pi$-m.p.). Furthermore, $\pi$ is the unique invariant distribution of the Markov kernel $P$ induced by the IRF.*

---

[3]For differentiable $g$ on continuous state spaces (e.g., $\mathbb{R}^d$), $J_g(x, v) = |\det \nabla g(x, v)|$ is its Jacobian determinant. We adopt the measure-theoretic formulation of Tierney [51] to handle arbitrary state spaces.

Assumption 2.2 implies that the sequence of iterates $X_t$ produced by the IRF behave like a $\pi$-invariant, irreducible Markov chain. Therefore, long-run averages of IRF iterates converge to expectations under $\pi$, following the standard law of large numbers (LLN) for MCMC [56], also known as the *random Birkhoff ergodic theorem* in the IRF literature [57; 58, Cor. 2.2.]. Theorem 2.3 synthesizes these results under Assumption 2.2, providing a unified statement for convenient use in our framework; proof can be found in Appendix D.1.

**Theorem 2.3.** *Suppose that IRF $f_\theta$ satisfies Assumption 2.2. Then, given $\phi \in L_1(\pi)$, we have that*

    *1. for $\pi$-a.e. $x \in \mathcal{X}$ and $\mu$-almost all $(\theta_t)_{t \in \mathbb{N}}$, as $T \to \infty$:*

$$\frac{1}{T} \sum_{t=0}^{T-1} \phi\left(f_{\theta_t} \circ \cdots \circ f_{\theta_1}(x)\right) \longrightarrow \mathbb{E}[\phi(X)], \quad X \sim \pi; \tag{5}$$

    *2. for $\mu$-almost all $(\theta_t)_{t \in \mathbb{N}}$, as $T \to \infty$:*

$$\frac{1}{T} \sum_{t=0}^{T-1} \phi\left(f_{\theta_t} \circ \cdots \circ f_{\theta_1}(x)\right) \xrightarrow{L^1(\pi)} \mathbb{E}[\phi(X)], \quad X \sim \pi. \tag{6}$$

*Moreover, the same results hold for the inverse IRF $\{f_\theta^{-1} : \theta \in \Theta\}$.*

## 3 Invertible measure-preserving IRF from involutive MCMC

In this section, we provide a concrete, general construction of invertible and exactly measure-preserving IRFs based on involutive MCMC kernels. The key idea, originally developed for the MH sampler [40, 45], is to further augment the space with two additional variables $u_v \in [0,1]^d, u_a \in [0,1]$. The variable $u_v$ pairs with the auxiliary variable $v$ of dimension $d$, and $u_a$ encodes the randomness in the accept/reject decision. Let the augmented target $\bar{\pi}$ and space $\mathcal{S}$ be defined as:

$$\bar{\pi}(s) = \pi(x)\rho(v \mid x)\mathbb{1}_{[0,1]^d}(u_v)\mathbb{1}_{[0,1]}(u_a), \quad s = (x, v, u_v, u_a) \in \mathcal{S} := \mathcal{X} \times \mathcal{V} \times [0,1]^d \times [0,1].$$

The two uniform auxiliary variables $u_v$ and $u_a$ will be refreshed with two random parameters $(\theta_v, \theta_a) \sim \mu = \text{Unif}[0,1]^d \times \text{Unif}[0,1]$. Without loss of generality, we describe the IRF construction assuming a one-dimensional target $\pi(x)$. The IRF $f_\theta(s) := f_\theta(x, v, u_v, u_a)$ is defined by the following steps (Algorithm 2):

    1. Uniform auxiliary refreshment: $u_v \leftarrow (u_v + \theta_v) \mod 1, \quad u_a \leftarrow (u_a + \theta_a) \mod 1$

    2. Update $(v, u_v)$ pair via CDF/inverse-CDF of $\rho(\cdot|x)$ [4]: $u_v' \leftarrow F_{\rho(\cdot|x)}(v), \quad \widetilde{v} \leftarrow F_{\rho(\cdot|x)}^{-1}(u_v)$

    3. Propose and compute the MH-ratio: $(x', v') \leftarrow g(x, \widetilde{v}), \quad r \leftarrow \frac{\bar{\pi}(x',v')}{\bar{\pi}(x,\widetilde{v})} J_g(x, \widetilde{v})$

    4. Accept or reject: If $u_a > r$, reject and stay at the *pre-involution* state $s' = (x, \widetilde{v}, u_v', u_a)$. Otherwise, set $u_a' \leftarrow \frac{u_a}{r}$ and accept the *post-involution* state $s' = (x', v', u_v', u_a')$.

The correspondence with involutive MCMC (Algorithm 1) is: Step 2 simulates $v \sim \rho(\mathrm{d}v|x)$ via inverse CDF sampling, Step 3 mirrors the involution and MH ratio computation, and Step 4 performs the accept/reject step while explicitly tracking the randomness $u_a$ involved in the decision. Furthermore, as mentioned in Section 4.1, one can use this map in a homogeneous MixFlow by simply fixing $\theta_v, \theta_a$ to some pre-specified constant values (rather than sampling from $\mu$). And finally, using the same Jacobian computation as in [45, Eq. (25)], one can show that $\forall \theta = (\theta_v, \theta_a) \in \Theta$, the IRF $f_\theta$ (Algorithm 2 of Appendix B) is $\bar{\pi}$-measure-preserving.

**Proposition 3.1.** *The map given by Algorithm 2 satisfies Assumption 2.2 for $\bar{\pi}$ if its induced Markov kernel $P$ is irreducible.*

One must be able to compute $f_\theta^{-1}(\cdot)$ if $f_\theta$ is to be used as a flow layer in a MixFlow. Steps 1–3 are straightforward to invert. The main challenge lies in inverting the accept/reject Step 4—we need to recover the accept/reject decision based solely on the output state $s'$. Depending on different decisions, $s'$ could either be the pre-involution state $(x, \tilde{v}, u_v', u_a)$ or the post-involution state $(x', v', u_v', u_a')$. Since the transformation $u_a' \leftarrow u_a/r$ only present in the acceptance branch, inferring the branch incorrectly would lead to the failure of recovering $u_a$ (hence the entire state $s$).

---

[4]Typically, $\rho(v|x)$ lies in a simple family; for instance, in HMC with a diagonal mass matrix, $\rho$ is a diagonal Gaussian, whose CDF and inverse-CDF can be computed stably. For multidimensional $v$, a Gibbs-style update on the conditionals of $\rho(\cdot|x)$ can be used.

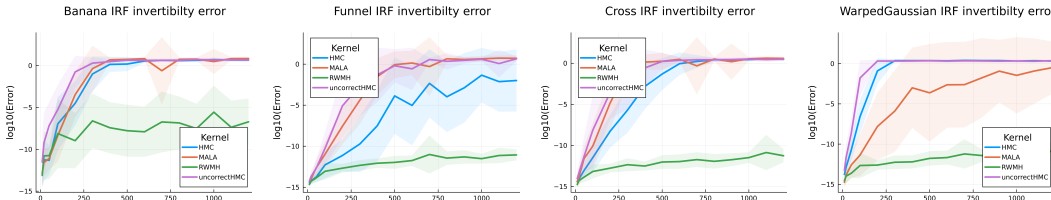

Figure 1: Inversion error of IRFs (based on HMC, uncorrected HMC, MALA, and RWMH) over increasing flow length $T$. Verticle axis shows the 2-norm error of reconstructing $s = (x, v, u_v, u_a)$ ($s = (x, v)$ for the uncorrected HMC IRF) sampled from $q_0$ by the composing the forward simulation $f_{\theta_T} \circ \cdots \circ f_{\theta_1}(s)$ and its inverse. The lines indicate the mean, and error regions indicate the standard deviation over 32 independent initializations from $q_0$.

We address this challenge (pseudocode in Algorithm 3 of Appendix B) by exploiting the self-inverse property of the involution $g$. First note that $g(x, \widetilde{v}) = (x', v')$ and $g(x', v') = (x, \widetilde{v})$. Suppose that $s' = (x^\#, v^\#, u_v^\#, u_a^\#)$. $\{g(x^\#, v^\#), (x^\#, v^\#)\}$ is exactly the unordered pair $\{(x, \widetilde{v}), (x', v')\}$. Then from the property of the Jacobian of $g$ (i.e., $J_g(x, \widetilde{v}) = J_g^{-1}(x', v')$ and vice versa), we observe

$$\frac{\pi(x', v')}{\pi(x, \widetilde{v})} J_g(x, \widetilde{v}) = \left( \frac{\pi(x, \widetilde{v})}{\pi(x', v')} J_g(x', v') \right)^{-1}.$$

Hence, recomputing the MH-ratio as in Step 3 yields

$$\widetilde{r} := \frac{\pi(x^\#, v^\#)}{\pi\left(g(x^\#, v^\#)\right)} \cdot J_g\left(g(x^\#, v^\#)\right) \in \{r, r^{-1}\},$$

where $r$ corresponds to the true MH-ratio as computed in the forward pass. The key observation to infer the accept/reject decision then follows: If $u_a^\# \cdot \widetilde{r} < 1$, then the acceptance branch was taken, so $u_a = u_a^\# \cdot \widetilde{r}$; otherwise the move was rejected as $u_a$ cannot be larger than 1.

Fig. 1 empirically verifies that one can successfully invert the proposed IRF map for four MCMC-based IRFs—HMC[59, 60], uncorrected HMC [61], MALA[62], and RWMH [63]—on four synthetic targets defined in Appendix E. The same hyperparameters are used for every example: each (uncorrected) HMC transition consists of 50 leapfrog steps with step size 0.02; MALA uses step size 0.25; RWMH uses step size 0.3. We evaluate the 2-norm error of reconstructing $s = (x, v, u_v, u_a)$ sampled from a mean-field Gaussian variational approximation $q_0$ by the composing the forward simulation $f_{\theta_T} \circ \cdots \circ f_{\theta_1}(s)$ and its inverse. Both HMC variants and MALA remain reliably invertible up to $T \approx 200$ iterations, while the RWMH IRF remain invertible up to $T \approx 1000$ iterations. Notably, the corrected HMC IRF is consistently more stable than its uncorrected counterpart used in past MixFlows work; the additional MH step discards trajectories with large numerical error that would otherwise cause the dynamics to diverge. Although Algorithm 2 and Algorithm 3 are exact inverses in theory, floating-point round-off accumulates with $T$ and exact reconstruction can fail [39, 50]. In practice, however, the resulting statistical error in downstream variational inference is often negligible, thanks to the *shadowing* property of chaotic dynamical systems [50].

## 4 Variational flows based on IRFs

In this section, we present a methodology that transforms any IRF system satisfying Assumption 2.2 into an *asymptotically exact* variational family. Alongside the exact homogeneous MixFlows derived from IRFs and their refined analysis, we introduce three additional families—each constructed from the same IRF but combined differently—and show that all converge to the target in total variation. Proofs are deferred to Appendix D. For simplicity, we present the methodology and theory using IRFs defined directly on the original space $\mathcal{X}$ rather than the augmented space $\mathcal{S}$. We also assume access to a reference distribution $q_0$ supporting i.i.d. sampling and tractable density evaluation.

### 4.1 Improved homogeneous MixFlows

As reviewed in Section 2.1, the homogeneous MixFlow $\bar{q}_T$ is defined as $\bar{q}_T = \frac{1}{T} \sum_{t=1}^{T} f^t q_0$ with the convention $\bar{q}_0 = q_0$. Given an IRF $f_\theta$ satisfying Assumption 2.2, one can construct a homogeneous flow map $f$ by fixing the parameter $\theta$ to a constant value $\theta^\star$ (e.g., $\pi/16$), rather than sampling

from the distribution $\mu$. This provides a generic way of building exact $\pi$-m.p. flow maps. A key property not noted in prior MixFlow work [39, 41] is a simplified expression for the density of $\bar{q}_T$ on arbitrary state spaces, enabled by a measure-theoretic formulation of the pushforward density under a $\pi$-m.p. bijection $f$, $fq_0(x) = \pi(x)\frac{q_0}{\pi}(f^{-1}x)$[5], as introduced in Appendix C. This yields a simplified form for the density of $\bar{q}_T$ (in contrast to Eq. (1)):

$$\bar{q}_T(x) = \frac{1}{T}\sum_{t=1}^{T} f^t q_0(x) = \pi(x) \cdot \frac{1}{T}\sum_{t=1}^{T} \frac{q_0}{\pi}(f^{-t}(x)), \quad \forall x \in \mathcal{X}. \tag{7}$$

In practice, this expression is particulary useful for evaluating the flow density; practitioners can evaluate $\bar{q}_T(x)$ without tracking the Jacobians of $f$ explicitly, which simplifies implementation and avoids numerical instability from accumulating Jacobians over long trajectories.

Moreover, the explicit expression Eq. (7) offers an intuitive understanding of why $\bar{q}_T$ converges. While the original convergence result in [39, Theorem 4.2] relied on general operator theory for e.m.p. systems [64], the density-based perspective is more transparent. If $f$ is $\pi$-e.m.p, the Birkhoff ergodic theorem [65; 64, p. 212] implies that $\frac{1}{T}\sum_{t=1}^{T} \frac{q_0}{\pi}(f^{-t}(x)) \to 1$. Consequently, for $\pi$-a.e. $x \in \mathcal{X}$, $\bar{q}_T(x) \to \pi(x)$ as $T \to \infty$. This enables a substantially simplified proof of the convergence of homogeneous MixFlow. The proof of Theorem 4.1 can be found in Appendix D.2.

**Theorem 4.1.** *Suppose that $f$ is a $\pi$-e.m.p diffeomorphism, and $q_0 \ll \pi$. Then, as $T \to \infty$,*
$$\bar{q}_T(x) \to \pi(x), \quad \pi\text{-a.e. } x \in \mathcal{X}, \quad and \quad \mathrm{TV}(\bar{q}_T, \pi) \to 0.$$

It is worth noting that Assumption 2.2 does not guarantee the ergodicity of a specific $f_{\theta^*}$, leaving a gap between theory and the practical implementation of homogeneous MixFlows. In the remainder of this section, we introduce three new MixFlow families designed to address this limitation.

## 4.2 IRF MixFlows

An IRF MixFlow is a mixture of pushforwards of a reference $q_0$ through an IRF sequence:
$$\overrightarrow{q_T} := \frac{1}{T}\sum_{t=1}^{T} f_{\theta_t} \circ \cdots \circ f_{\theta_1} q_0, \quad \text{with the convention that } \overrightarrow{q_0} = q_0,$$

where $\theta_1, \ldots, \theta_T$ is a *cached* i.i.d. sequence drawn from $\mu$. When constructing the flow, we first sample and freeze the random stream $\theta_1, \ldots, \theta_T$, yielding an *inhomogeneous* sequence of $T$ parameterized bijections. Then to draw $X \sim \overrightarrow{q_T}$, we treat $\overrightarrow{q_T}$ as a mixture of $T$ distributions:
$$K \sim \mathrm{Unif}\{1, 2, \ldots, T\} \qquad X_0 \sim q_0 \qquad X = f_{\theta_K} \circ \cdots \circ f_{\theta_1}(X_0)$$
Note crucially that each sample $X$ is generated using the *same frozen* sequence $\theta_1, \ldots, \theta_T$. For density evaluation, we compute the inverse IRF $f_{\theta_T}^{-1}, \cdots, f_{\theta_1}^{-1}$. Because each $f_\theta$ is $\pi$-m.p., by Proposition C.3, the density takes a similar form as in a homogeneous MixFlow (Eq. (7)):

$$\overrightarrow{q_T}(x) = \pi(x) \cdot \frac{1}{T}\sum_{t=1}^{T} \frac{q_0}{\pi}\left(f_{\theta_1}^{-1} \circ \cdots \circ f_{\theta_t}^{-1}(x)\right), \quad \forall x \in \mathcal{X}.$$

However, note that this density requires simulating the *backward process* of the inverse IRF ([47])
$$\overleftarrow{X_t}(x) := f_{\theta_1}^{-1} \circ \cdots \circ f_{\theta_t}^{-1}(x) \quad \text{for } t \in [T],$$
which cannot be computed sequentially. As a result, IRF MixFlows incur a quadratic density evaluation cost $O(T^2)$. Fortunately, this backward process can be computed in a parallel fashion, as the computation of each $\overleftarrow{X_t}(x)$, $t \in [T]$ is independent. We recommend deploying IRF MixFlows on modern parallel hardware (e.g., GPUs) for efficient density evaluation.

IRF MixFlows share the total variation convergence guarantee (Theorem 4.2) of homogeneous MixFlows. The proof (Appendix D.3.1) is similar to the original MixFlow argument [39, Theorem 4.2], interpreting the IRF (Eq. (2)) as a time-homogeneous, e.m.p. dynamical system over the joint space $\Theta^\mathbb{N} \times \mathcal{X}$. However, we emphasize that Assumption 2.2 is significantly weaker than the ergodicity assumption of Theorem 4.1. See Section 4.5 for a detailed discussion.

**Theorem 4.2.** *Let $\mathbb{P}$ denote the joint distribution over the i.i.d. sequence $(\theta_t)_{t\in\mathbb{N}} \overset{iid}{\sim} \mu$. If Assumption 2.2 holds and $q_0 \ll \pi$, then*
$$\mathrm{TV}\left(\overrightarrow{q_T}, \pi\right) \overset{\mathbb{P}}{\longrightarrow} 0 \quad as \ T \to \infty. \tag{8}$$

---

[5]An implication of this result in continuous state space is that for any $\pi$-m.p. diffeomorphism $f$, the Jacobian determinant must satisfy $|\det \nabla f^{-1}(x)| = \frac{\pi(f^{-1}x)}{\pi(x)}$, as established in Proposition C.3.

## 4.3 Backward IRF MixFlows

To address the $O(T^2)$ density cost of IRF MixFlows, we propose a simple modification: constructing the flow from the *backward process*. Specifically, we define the *backward IRF MixFlow* as:

$$\overleftarrow{q_T} := \frac{1}{T} \sum_{t=1}^{T} f_{\theta_1} \circ \cdots \circ f_{\theta_t} q_0, \quad \text{with the same convention that } \overleftarrow{q_0} = q_0.$$

This construction retains $O(T)$ complexity of sampling $X \sim \overleftarrow{q_T}$ via:

$$K \sim \text{Unif}\{1, 2, \ldots, T\} \qquad X_0 \sim q_0 \qquad X = f_{\theta_1} \circ \cdots \circ f_{\theta_K}(X_0),$$

while reducing the density computation cost to $O(T)$. The density of $\overleftarrow{q_T}$ is given by:

$$\overleftarrow{q_T}(x) = \pi(x) \cdot \frac{1}{T} \sum_{t=1}^{T} \frac{q_0}{\pi} \left( f_{\theta_t}^{-1} \circ \cdots \circ f_{\theta_1}^{-1}(x) \right), \quad \forall x \in \mathcal{X}. \tag{9}$$

This mirrors the density formula of homogeneous MixFlows (Eq. (7)), enabling the use of the random ergodic theorem (Theorem 2.3) to establish the same pointwise and total variation convergence.

**Theorem 4.3.** *If Assumption 2.2 holds and $q_0 \ll \pi$, then for $\pi$-a.e. $x \in \mathcal{X}$ and $\mu$-almost all $(\theta_t)_{t \in \mathbb{N}}$:*

$$\overleftarrow{q_T}(x) \longrightarrow \pi(x) \quad and \quad \text{TV}(\overleftarrow{q_T}, \pi) \longrightarrow 0 \quad as \ T \to \infty.$$

## 4.4 Ensemble IRF MixFlows

All MixFlow variants discussed above—including homogeneous MixFlows—are based on *ergodic averaging* along the flow. This inherently limits their convergence rate to $O(1/T)$, as the first component always retains a $1/T$ mixing weight. In contrast, MCMC methods often exhibit geometric convergence in their marginal distributions under suitable conditions [56; 66, Ch. 15]. Motivated by this, we propose the *ensemble IRF MixFlows*, which instead uses an *ensemble average* of the endpoint of multiple IRF trajectories in an attempt to match $T$-step MCMC marginal distribution:

$$\widetilde{q}_T^{(M)} := \frac{1}{M} \sum_{m=1}^{M} q_T^{(m)} = \frac{1}{M} \sum_{m=1}^{M} f_{\theta_T^{(m)}} \circ \cdots \circ f_{\theta_1^{(m)}} q_0,$$

where each $\theta_1^{(m)}, \ldots, \theta_T^{(m)}$ corresponds to an independent IRF realization. As in the case of the previous MixFlows, the $M$ streams of randomness $\theta_t^{(m)}$ are cached (i.e., *frozen*) when sampling and computing densities. The resulting density of the ensemble IRF MixFlow is given by:

$$\widetilde{q}_T^{(M)}(x) = \pi(x) \cdot \frac{1}{M} \sum_{m=1}^{M} \frac{q_0}{\pi} \left( f_{\theta_1^{(m)}}^{-1} \circ \cdots \circ f_{\theta_T^{(m)}}^{-1}(x) \right),$$

whose computation costs $O(TM)$ (or $O(T)$ when parallelized across the $M$ streams). Drawing $X \sim \widetilde{q}_T^{(M)}$ takes $O(T + M)$ operations:

$$K \sim \text{Unif}\{1, 2, \ldots, M\} \qquad X_0 \sim q_0 \qquad X = f_{\theta_T^{(K)}} \circ \cdots \circ f_{\theta_1^{(K)}} q_0.$$

Intuitively, the flow length $T$ controls the bias of the IRF system, while the ensemble size $M$ controls the variance of the Monte Carlo average. This tradeoff is formalized in the following result.

**Theorem 4.4.** *Suppose that Assumption 2.2 holds, and that $\forall x \in \mathcal{X}, \frac{q_0}{\pi}(x) \leq B < \infty$. Then,*

$$\mathbb{E}_\theta \left[ \text{TV} \left( \widetilde{q}_T^{(M)}, \pi \right) \right] \leq \frac{1}{\sqrt{M}} \mathbb{E} \left[ \sqrt{\text{Var}_{\theta_{1:T}} \left[ \frac{q_0}{\pi} \left( f_{\theta_1}^{-1} \circ \cdots \circ f_{\theta_T}^{-1}(X) \right) \mid X \right]} \right]$$
$$+ B \cdot \mathbb{E} \left[ \text{TV}(R^T \delta_X, \pi) \right], \quad X \sim \pi,$$

*where $R$ is the Markov kernel induced by the inverse IRF $f_\theta^{-1}$, and $\delta_X$ is the Dirac measure at $X$.*

In the setting where $\text{TV}(R^T \delta_X, \pi) = O(\rho^T)$ for some $\rho \in (0, 1)$, the convergence rate of $\widetilde{q}_T^{(M)}$ can be heuristically characterized as $\text{TV}\left( \widetilde{q}_T^{(M)}, \pi \right) = O\left( \rho^T \vee \frac{1}{\sqrt{M}} \right)$, capturing the tradeoff between the bias (via $T$) and variance (via $M$). Given a fixed computational budget, choosing the balance between flow length $T$ and ensemble size $M$ is critical. In the extreme case of $M = 1$, convergence will fail entirely—any $\pi$-measure-preserving $f$ satisfies $\text{TV}(fq, \pi) = \text{TV}(q, \pi)$ [67, Theorem 1]. On the other hand, small $T$ leads to high bias due to insufficient mixing. This tradeoff closely relates to recent studies on parallel MCMC algorithms [68, 69].

### 4.5 Discussion

**Relaxing ergodicity.** A major advantage of IRF-based MixFlows over homogeneous MixFlows is that IRF-based MixFlows require only that the kernel $P$ admits a unique invariant distribution (Assumption 2.2), a significantly weaker condition than the ergodicity assumed by homogeneous MixFlows. In fact, whenever the set $\Theta^\star := \{\theta : f_\theta \text{ is } \pi\text{-ergodic}\}$ has positive $\mu$-measure, Assumption 2.2 automatically holds [46, Corollary 3.3]. Uniqueness of the invariant distribution is also easily verified by checking that $P$ is irreducible [56, 66]. The IRFs we construct in Section 3 correspond to involutive MCMC kernels that are known to be irreducible, whereas establishing ergodicity in MixFlows is typically so difficult that it is assumed without proof [39, 42, 70].

**Which flow to choose?** All four flows are asymptotically exact, yet their density formulae reveal different bias-variance and cost-accuracy trade-offs. In every case the density ratio takes the form $\frac{\text{flow density}}{\pi}(x) = \frac{1}{N}\sum_{n=1}^{N}\frac{q_0}{\pi}(T_n(x))$, where $T_n$ is a composition of inverse IRF/ergodic maps, and $N$ can be the flow length or ensemble size. Hence practical convergence of each flow is dictated by how quickly $\frac{1}{N}\sum_{n=1}^{N}\frac{q_0}{\pi}(T_n(x))$ converges to a constant. Empirically (see Appendix E.1.1) we find that IRF MixFlows often reach a given accuracy at shorter flow lengths than homogeneous or backward IRF MixFlows, but a full theoretical comparison study is deferred to future work.

## 5 Experiments

This section presents an empirical evaluation of the four proposed flows—three IRF variants and homogeneous MixFlows (collectively referred to as "IRF flows" since homogeneous MixFlows can be viewed as a special case). We compare them against two normalizing flows, RealNVP [19] and Neural Spline Flow (NSF) [71], and against the No-U-Turn Sampler (NUTS) [72]. Variational methods are assessed by their (i) ELBO and (ii) accuracy of the importance sampling estimate of the normalization constant $\log Z$ for the unnormalized density $\gamma$:

$$Z \approx \frac{1}{N}\sum_{n=1}^{N}\frac{\gamma}{q_T}(X_n), \quad (X_n)_{n=1}^{N} \overset{\text{iid}}{\sim} q_T, \quad \text{where } q_T \in \left\{\bar{q}_T, \overrightarrow{q}_T, \overleftarrow{q}_T, \widetilde{q}_T^{(M)}\right\}, \pi = \frac{\gamma}{Z}$$

and (iii) importance sampling effective sample size (ESS) [73–75]. Sampling methods are evaluated via their Monte Carlo estimation error. In all cases, all flows start from the same reference distribution $q_0$: a mean-field Gaussian trained for 10K Adam steps with batch size 10 and learning rate $10^{-3}$. All IRF flows are evaluated with 64 i.i.d. draws, while normalizing flows use 1024. Full experimental details appear in Appendix E.

### 5.1 Synthetic examples

Our synthetic experiments consist of four 2-dimensional targets used by Xu et al. [39]: the Banana [76], Neal's funnel [77], a cross-shaped Gaussian mixture, and a warped Gaussian distribution. Fig. 2 shows a comparison of the original Hamiltonian-MixFlow—built on an *uncorrected* HMC kernel—with our *corrected* version including the MH step. For each target we run both flows with identical hyper-parameters (50 leapfrog steps per transition, several step-sizes) and estimate the total-variation (TV) distance to the ground truth using 512 i.i.d. samples. Across all targets and step-sizes, the corrected HMC-based MixFlow consistently achieves lower TV error and remains robust as the step-size grows. In contrast, the uncorrected variant often deteriorates with longer flows because the inexact map error accumulates (e.g., the green dashed curve in the third panel). At larger step sizes the uncorrected flow frequently diverges, producing NaNs (marked by crosses), whereas the corrected flow remains stable—echoing the inversion stability results in Fig. 1.

We next compare the four IRF flows with `RealNVP` and `NSF`. Two IRF variants are examined: HMC-based (50 leapfrog steps per transition; $T = 200$) and RWMH-based ($T = 4000$). Each normalizing flow consists 6 flow layers, and is trained via 50,000 Adam steps with batch size 32; we tune the learning rates in the grid $\{10^{-4}, 10^{-3}, 10^{-2}\}$, and report the results of the setting with smallest median TV distance over 5 runs. Additional implementation details can be found in Appendix E.1.

Figs. 3a and 3b display the ELBO and $\log Z$ estimates (via importance sampling) for the Banana target; the remaining synthetic cases show the same pattern (Fig. 7 in Appendix E.1.3). As synthetic targets are normalized, a perfect variational approximation has both metrics near 0. The IRF flows meet this mark consistently across runs, whereas `RealNVP` and `NSF` exhibit high variability and often

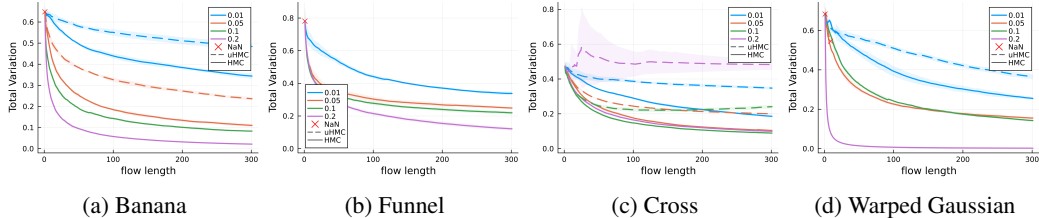

| (a) Banana | (b) Funnel | (c) Cross | (d) Warped Gaussian |

Figure 2: Total-variation error for homogeneous MixFlow built on *corrected* (solid) versus *uncorrected* (dashed) HMC kernels, plotted against flow length $T$ for several step sizes. Each curve is the mean over 32 independent runs; shaded bands ($\pm 1$ SD) show run-to-run variability. A cross marks any setting where at least one run returned a `NaN` (instability), at which point the trace is terminated.

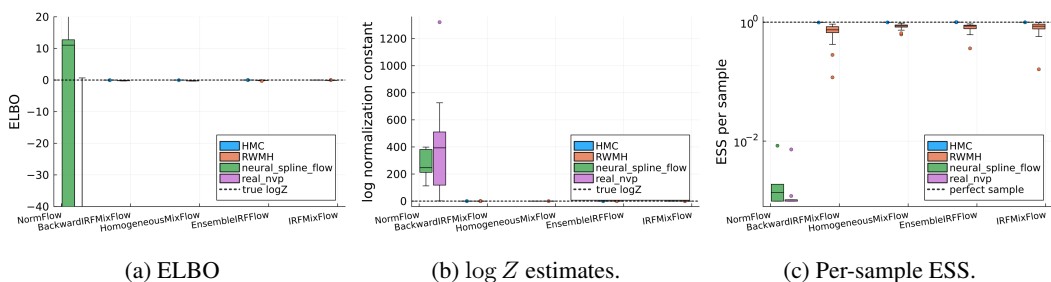

| (a) ELBO | (b) $\log Z$ estimates. | (c) Per-sample ESS. |

Figure 3: Variational approximation quality of IRF Flows versus `RealNVP` and `NSF`. Box plots for IRF flows are based on 32 independent runs, and 10 runs for the normalizing flows. The black dashed line in (c) indicates the optimal ESS of perfect i.i.d. samples.

produce extreme ELBO or $\log Z$ values. We restrict the vertical range of the ELBO plot for better visualization; full-range plots are in Fig. 7b. We also note that training instability is common for the normalizing flows: on the Funnel example, 10 of 15 `RealNVP` runs and all `NSF` runs diverged.

Fig. 3c further examine the per-sample importance sampling ESS (see Fig. 7d on similar results for other examples), which reflects the $\chi^2$ divergence from the variational distribution to the target [78]. The ESS is orders of magnitude higher for IRF flows than for the normalizing flows. Additionally, we provide comparisons among the three ergodic averaging MixFlow variants in Appendix E.1.1, and ensemble-size/length trade-offs for ensemble IRF MixFlows are explored in Appendix E.1.2.

## 5.2 Real-data experiments

The real-data experiments include the Student-t-regression (`TReg`; 4-dimensional), and the Sparse linear regression (`SparseReg`; 83-dimensional) from [39], and a latent Brownian motion model (`Brownian`; 32-dimensional) and the Log-Gaussian Cox process model (`LGCP`; 1600-dimensional) from the Inference Gym library [79]. Each normalizing flow is trained via $50{,}000$ Adam steps of batch size 32; we grid-search both the learning rates $\{10^{-4}, 10^{-3}, 10^{-2}\}$ and flow layers $\{6, 10\}$, and report the configuration with the highest median ELBO over 5 runs. An additional mean-field Gaussian baseline is optimized for the same number of steps and batch size with learning rate $10^{-3}$.

All IRF variants use RWMH kernel, with the step size tuned to achieve a $0.8$ acceptance rate using bisection search between $0.001$ and $10$. In each search step, we estimate acceptance rate with $5{,}000$ RWMH-IRF iterations. We set $T = 5000$ for the backward IRF and homogeneous MixFlow and ensemble IRF MixFlow, and set $T = 4000$ for the IRF MixFlow. Normalizing flow results are omitted for `LGCP`, which did not finish training within 48 hours on the same computation cluster. Ground truth values are estimated using AIS with a dense temperature grid; see the details in Appendix E.2.

As in the synthetic experiments, our exact flows match—or modestly improve upon—the best-tuned `RealNVP` and `NSF` in both ELBO (Fig. 4a) and $\log Z$ accuracy (Fig. 4b), and outperform the mean-field baseline by a wide margin. The per-sample importance-sampling ESS shows the same advantage (Fig. 8b). Crucially, normalizing flow training is orders of magnitude more expensive (Fig. 4d), whereas the exact flows achieve comparable accuracy at a fraction of the computational cost.

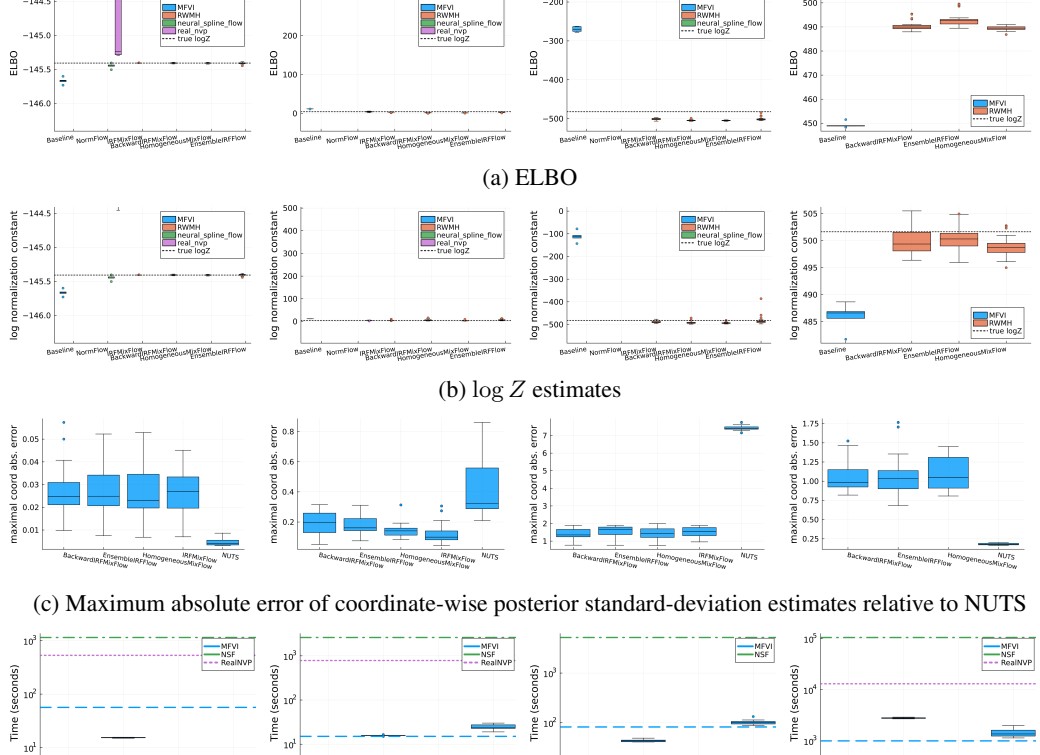

(a) ELBO

(b) $\log Z$ estimates

(c) Maximum absolute error of coordinate-wise posterior standard-deviation estimates relative to NUTS

(d) Computation time (in seconds) for each method. To ensure a consistent environment, all timing results were obtained by rerunning the methods on the same local machine (hardware details provided in Appendix E). MFVI, NSF, and RealNVP were each run once, as their execution times are deterministic given the flow architecture and optimization settings. For IRF flows and NUTS, timing statistics are based on 10 independent runs.

Figure 4: Results on real-data benchmarks (columns, from left to right): TReg($d = 4$), Brownian($d = 32$), SparseReg ($d = 83$), and LGCP ($d = 1600$).

We further compare coordinate-wise posterior mean estimates (Fig. 8c in Appendix E.2) and standard deviation estimates (Fig. 4c) against NUTS, reporting the maximum absolute error across dimensions relative to the estimated ground truth. NUTS is initialized with independent draws from $q_0$ and run for 10,000 iterations including 5000 warm-up iterations. IRF flows outperform NUTS on two models and are slightly worse on the other two—yet they do so at generally faster computation time (Fig. 4d). Note that the goal of this work is not to outperform MCMC, but rather to construct a variational family that provides asymptotic exactness and similar sampling performance; IRF MixFlows meet this standard.

## 6 Conclusion

We introduced a general framework for building asymptotically exact variational families from general involutive MCMC kernels. By constructing invertible, measure-preserving maps directly from these kernels, we overcome the main practical limitation of MixFlow [39] and enable the construction of a broad class of exact flows. We also provided a streamlined theoretical analysis for flows based on measure-preserving transformations and demonstrated their empirical advantages in density approximation and importance sampling. A promising direction is to pair our framework with recent automatic-tuning MCMC [52–55], developing truly tuning-free exact flows in practice.

## Acknowledgments and Disclosure of Funding

The authors sincerely thank Peter Orbanz for pointing us to the IRF literature, which provided the initial inspiration for this work, and Alexandre Bouchard-Côté for suggesting applying our methods to normalizing constant estimation. T. Campbell and Z. Xu acknowledge support from the NSERC Discovery Grant RGPIN-2025-04208. We are also grateful for access to the ARC Sockeye computing platform at the University of British Columbia, and the compute cluster provided by the Digital Research Alliance of Canada.

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

# Contents

# A  Additional content about involutive MCMC

## A.1  Examples of involutive MCMC

Here, we illustrate how the generic Metropolis-Hastings (MH) algorithm [48, 51], random-walk Metropolis-Hastings (RWMH) [63, 80], and Hamiltonian Monte Carlo (HMC) [59, 60], fit into this framework by specifying the corresponding auxiliary distribution $\rho(\cdot|x)$ and the involution map $g$.

**Example A.1** (MH sampler; Section B.3. of [49])**.** *The Metropolis-Hastings sampler with proposal distribution $\rho(\mathrm{d}x'|x)$ can be cast as an involutive MCMC method by defining the auxiliary distribution as $\rho(\mathrm{d}v|x)$, and using the swap involution $g : (x, v) \mapsto (v, x)$.*

**Example A.2** (RWMH sampler; Section 2. of [52])**.** *RWMH with step size $\epsilon$ is obtained by setting*

$$g(x, v) = (x + \epsilon v, -v), \quad v \sim \rho(\mathrm{d}v|x) = \mathcal{N}(0, I).$$

**Example A.3** (HMC; [60])**.** *In the involutive formulation of HMC, the auxiliary variable $v$ corresponds to the momentum variable, and $\rho(v|x)$ is the momentum distribution, typically a Gaussian distribution independent of $x$. The involution map $g$ consists of applying $k$ steps of the leapfrog integrator, followed by a momentum sign flip:*

$$g\left(\begin{bmatrix} x \\ v \end{bmatrix}\right) = \begin{bmatrix} I & 0 \\ 0 & -I \end{bmatrix} L^k\left(\begin{bmatrix} x \\ v \end{bmatrix}\right),$$

*where $L : (x, v) \to (x', v')$ denotes a single leapfrog step (of step size $\epsilon$) given by*

$$v_{1/2} \leftarrow v + \frac{\epsilon}{2}\nabla \log \pi(x)$$
$$x' \leftarrow x + \epsilon v_{1/2}$$
$$v' \leftarrow v_{1/2} + \frac{\epsilon}{2}\nabla \log \pi\left(x'\right).$$

## A.2  Pseudocode of involutive MCMC

---
**Algorithm 1** Involutive MCMC kernel $K(x', v'|x, v)$

---
**Require:** current state $x$, target $\pi$, auxiliary distribution $\rho(\mathrm{d}v|x)$, involution $g$
 1: $v \sim \rho(\mathrm{d}v|x)$          ▷ sample auxiliary variable
 2: $(x', v') \leftarrow g(x, v)$          ▷ generate proposal via the involution
 3: $\alpha \leftarrow \min\left(1, \frac{\overline{\pi}(x', v')}{\overline{\pi}(x, v)} J_g(x, v)\right)$          ▷ compute the acceptance probability
     ▷ Accept or reject
 4: $u \sim \mathrm{Unif}[0, 1]$
 5: **if** $u > \alpha$ **then**
 6:     $x' \leftarrow x$          ▷ reject
 7: **end if**
 8: **return** $x', v'$

---

# B    Pseudocode for IRF and inverse IRF based on involutive MCMC

---

**Algorithm 2** IRF based on involutive MCMC $f_\theta(s)$

---

**Require:** joint state $s = (x, v, u_v, u_a)$, random parameters $\theta = (\theta_v, \theta_a)$

    ▷ update uniform auxiliary variables

1: $u_v \leftarrow (u_v + \theta_v) \mod 1$

2: $u_a \leftarrow (u_a + \theta_a) \mod 1$

    ▷ involutive MCMC with target $\pi(x)$, auxiliary distribution $\rho(\mathrm{d}v|x)$, involution $g$

3: $u'_v \leftarrow F_{\rho(\cdot|x)}(v)$

4: $\widetilde{v} \leftarrow F^{-1}_{\rho(\cdot|x)}(u_v)$

5: $(x', v') \leftarrow g(x, \widetilde{v})$

6: $r \leftarrow \frac{\overline{\pi}(x', v')}{\overline{\pi}(x, \widetilde{v})} J_g(x, \widetilde{v})$                                    ▷ Compute MH ratio

7: **if** $u_a > r$ **then**

8:      **return** $x, \widetilde{v}, u'_v, u_a$                         ▷ reject and return pre-involution state

9: **end if**

10: $u'_a \leftarrow \frac{u_a}{r}$                             ▷ $u_a \leq r$ implies that $u'_a \in [0, 1]$

11: **return** $x', v', u'_v, u'_a$                    ▷ accept and return after-involution state

---

**Algorithm 3** Inverse IRF based on involutive MCMC $f_\theta^{-1}(s')$

---

**Require:** joint state $s' = (x', v', u'_v, u'_a)$, random parameters $\theta = (\theta_v, \theta_a)$

    ▷ recover pre- and post-involution pair

1: $(x, \widetilde{v}) \leftarrow g(x', v')$

    ▷ this will either be $r$ in line 6 of Algorithm 2 if accepted, or $r^{-1}$ otherwise

2: $\widetilde{r} \leftarrow \frac{\overline{\pi}(x', v')}{\overline{\pi}(x, \widetilde{v})} J_g(x, \widetilde{v})$

    ▷ check accept or reject

3: $u_a \leftarrow u'_a \cdot \widetilde{r}$        ▷ update $u_a$ (line 10 of Algorithm 2) as if the forward pass was an accept

4: **if** $u_a > 1$ **then**                    ▷ forward pass was a reject (see line 6-7 of Algorithm 2)

    ▷ pre-involution state

5:      $(x, \widetilde{v}) \leftarrow x', v'$

6:      $u_a \leftarrow u'_a$

7: **end if**

    ▷ inverse of line 3-4 of Algorithm 2

8: $v \leftarrow F^{-1}_{\rho(\cdot|x)}(u_v)$

9: $u_v \leftarrow F_{\rho(\cdot|x)}(\widetilde{v})$

    ▷ inverse update of the uniform auxiliary variables (line 1-2 of Algorithm 2)

10: $u_v \leftarrow (u_v + 1 - \theta_v) \mod 1$

11: $u_a \leftarrow (u_a + 1 - \theta_a) \mod 1$

12: **return** $x, v, u_v, u_a$

---

# C  Measure-theoretic formulation of pushforward density

A fundamental formula when studying variational inference is the the change of variable formula, which characterizes the density of a transformed distribution. For a diffeomorphism $f : \mathcal{X} \to \mathcal{X}$ on a continuous space, the density of $X = f(Y), Y \sim q_0$, is given by

$$\forall x \in \mathcal{X}, \quad q_\lambda(x) = fq_0(x) = \frac{q_0\left(f^{-1}(x)\right)}{J\left(f^{-1}(x)\right)}, \quad J(x) = |\det \nabla f(x)|\,.$$

However, the assumptions of differentiability and a continuous state space can be restrictive, as many inference problems involve discrete or hybrid spaces (e.g., Ising model [81], Bayesian Gaussian mixture model [82], and spike-and-slab model [83]). To handle general state spaces, we adopt a measure-theoretic formulation of the pushforward density, stated in Proposition C.1, for a generic bijection $f$. This result is well known (see, e.g., 51 for its use in the general involutive MCMC framework), but we include a proof here for completeness.

**Proposition C.1.** *Suppose that $f : \mathcal{X} \to \mathcal{X}$ is bijective. For a distribution $q \ll \pi$, for all $x \in \mathcal{X}$ :*

$$\frac{\mathrm{d}(fq)}{\mathrm{d}\pi}(x) = \frac{\mathrm{d}q}{\mathrm{d}\pi}\left(f^{-1}x\right)\frac{\mathrm{d}f\pi}{\mathrm{d}\pi}(x).$$

*Proof of Proposition C.1.* First, note that if $q \ll \pi$, then $fq \ll f\pi$. This implies that

$$\frac{\mathrm{d}(fq)}{\mathrm{d}\pi}(x) = \frac{\mathrm{d}(fq)}{\mathrm{d}\pi}(x)\frac{\mathrm{d}f\pi}{\mathrm{d}\pi}(x), \quad \forall x \in \mathcal{X}.$$

It remains to show that $\frac{\mathrm{d}(fq)}{\mathrm{d}f\pi} = \frac{\mathrm{d}q}{\mathrm{d}\pi} \circ f^{-1}$. It suffices to show that $\forall A \in \mathcal{B}$,

$$\int_A \frac{\mathrm{d}q}{\mathrm{d}\pi} \circ f^{-1}\mathrm{d}f\pi = \int_A \frac{\mathrm{d}(fq)}{\mathrm{d}f\pi}\mathrm{d}f\pi = fq(A).$$

Note that for all $A \in \mathcal{B}$, we have that

$$\int_A \frac{\mathrm{d}q}{\mathrm{d}\pi}(f^{-1}x)f\pi(\mathrm{d}x) = \int_{f^{-1}(A)} \frac{\mathrm{d}q}{\mathrm{d}\pi}(x)\pi(\mathrm{d}x) = q(f^{-1}A) = fq(A),$$

which completes the proof. $\qquad\square$

It is worth noting that for a Euclidean space $\mathcal{X}$ equipped with the Lebesgue measure $m$, and a diffeomorphism $f$, $\frac{\mathrm{d}fm}{\mathrm{d}m}(x)$ is precisely the Jacobian determinant $|\det \nabla f^{-1}(x)|$.

If $f$ is further $\pi$-measure-preserving, then $\frac{\mathrm{d}f\pi}{\mathrm{d}\pi} = 1$, yielding a simplified expression for the pushforward density.

**Corollary C.2.** *Suppose that $f$ is bijective and $\pi$-measure-preserving. For a distribution $q \ll \pi$, for all $x \in \mathcal{X}$:*

$$\frac{\mathrm{d}(fq)}{\mathrm{d}\pi}(x) = \frac{\mathrm{d}q}{\mathrm{d}\pi}(f^{-1}x).$$

Aside from the generality of Corollary C.2 over the diffeomorphic case, it provides an elegant formula of the pushforward density under a measure-preserving map. We invoke Corollary C.2 frequently when developing and analyzing MixFlows.

Beyond extending the diffeomorphic case, Corollary C.2 offers an elegant expression for the pushforward density under a measure-preserving map. We frequently invoke this result when developing and analyzing MixFlows. Finally, we present a specialization of Corollary C.2 for diffeomorphic $f$, which provides a convenient characterization of $\pi$-measure-preservation.

**Proposition C.3.** *Let $f : \mathcal{X} \to \mathcal{X}$ be a diffeomorphism, $\pi$ be a probability distribution on $\mathcal{X}$, with density (denoted by $\pi(x)$) with respect to a dominating measure $\lambda$. Then,*

1. *$f$ is $\pi$-measure-preserving if and only if $f^{-1}$ is $\pi$-measure-preserving.*

2. *$f$ is $\pi$-measure-preserving if and only if for $\lambda$-a.e. $x \in \mathcal{X}$, $J_f(x) := \left|\det \nabla f^{-1}(x)\right| = \frac{\pi(x)}{\pi(f^{-1}(x))}$.*

*Proof of Proposition C.3.* By definition, $f$ is $\pi$-preserving if and only if $f\pi(x) = \pi(x) = f^{-1}\pi(x)$. Examining the density of the pushforward $f\pi$ via the change-of-variable formula, we have

$$\forall x \in \mathcal{X}, \quad f\pi(x) = \pi(f^{-1}(x))J_f(x) = \pi(x) \Leftrightarrow J_f(x) = \frac{\pi(x)}{\pi(f^{-1}(x))}.$$

The second claim follows from the fact that $\pi = (f \circ f^{-1})\pi = (f^{-1} \circ f)\pi$. $\qquad\square$

# D  Proofs

## D.1  Proof of Theorem 2.3

As introduced in the main text, the IRF $f_\theta$ induces a Markov kernel given by:

$$\forall x \in \mathcal{X}, \quad \forall B \in \mathcal{B}, \quad P(x, B) := \int_\Theta \mathbb{1}_B(f_\theta(x))\mu(\mathrm{d}\theta).$$

This yields a simple characterization of the action of the Markov process $P$ on a distribution $q$:

$$(Pq)(y) := \int_\mathcal{X} P(x, y)q(\mathrm{d}x) = \mathbb{E}[f_\theta q(y)], \quad \theta \sim \mu, \quad f_\theta q: \text{pushforward of } q \text{ under } f_\theta.$$

We can further characterize the Markov kernel $R(\cdot, \cdot)$ induced by the *inverse IRF* $f_{\theta_t}^{-1}$:

$$\forall x \in \mathcal{X}, \quad \forall A \in \mathcal{B}, \quad R(x, A) := \int_\Theta \mathbb{1}_A(f_\theta^{-1}(x))\mu(\mathrm{d}\theta).$$

which is precisely the *reversal* of $P(\cdot, \cdot)$:

$$\pi \otimes P(A \times B) = \pi \otimes R(B \times A) = \int \pi(f_\theta(A) \cap B)\mu(\mathrm{d}\theta), \tag{10}$$

where $\pi \otimes P(A \times B) := \int_A P(x, B)\pi(\mathrm{d}x)$. See Kakutani [57, Eq. (4.5)] for the detailed derivation. Notice that if $P$ is reversible wrt $\pi$, i.e., $\pi \otimes P = \pi \otimes R$, both the IRF $f_\theta$ and its inverse $f_\theta^{-1}$ induce the same Markov process $P$. In other words, $P = R$. From Eq. (10), we can see that a sufficient and necessary condition so that $P = Q$ is that

$$\int \pi(f_\theta(A) \cap B)\mu(\mathrm{d}\theta) = \int \pi(f_\theta^{-1}(A) \cap B)\mu(\mathrm{d}\theta).$$

*Proof of Theorem 2.3.* From Eq. (4), we see that $P$ must admit $\pi$ as a stationary distribution. Douc et al. [66, Theorem 5.2.6] further states that if $\pi$ is the unique invariant probability measure of $P$, then the Markov process $P$ is ergodic. Therefore, the LLM of ergodic Markov process [66, Theorem 5.29] guarantees Eq. (5), and the random ergodic theorem [58, Cor. 2.2.] ensures Eq. (6).

Then as discussed above, Kakutani [57, Theorem 3.] show that Assumption 2.2 holds for $f_\theta$ and its induced Markov process $P$ if and only if Assumption 2.2 holds for the inverse IRF $f_\theta^{-1}$ and its induced $R$. Therefore, the same convergence holds for the inverse IRF. $\quad\square$

## D.2  Convergence of the homogeneous MixFlow

**Definition D.1** (Ergodic map [64, pp. 73, 105]). *$f : \mathcal{X} \to \mathcal{X}$ is ergodic for $\pi$ if for all measurable sets $A \subseteq \mathcal{X}$, $f(A) = A$ implies that $\pi(A) \in \{0, 1\}$.*

The most notable implication of a $\pi$-e.m.p $f$ is that the long-run average of repeated applications of $f$ converges to the expectation under $\pi$, a result known as the Birkhoff ergodic theorem [65; 64, p. 212]. The full statement is given in Theorem D.2.

**Theorem D.2** (Ergodic Theorem [65; 64, p. 212]). *Suppose $f : \mathcal{X} \to \mathcal{X}$ is measure-preserving and ergodic for $\pi$, and $\phi \in L^1(\pi)$. Then*

$$\lim_{T \to \infty} \frac{1}{T} \sum_{t=1}^T \phi(f^t x) = \int \phi \mathrm{d}\pi, \qquad \pi\text{-a.e. } x \in \mathcal{X}.$$

**Lemma D.3** (Scheffé's Lemma). *Let $\phi_n$ be a sequence of integrable functions on a measure space $(\mathcal{X}, \mathcal{B}, \pi)$ that convergences $\pi$-a.s. to $\phi$. Then*

$$\int |\phi_n(x) - \phi(x)|\pi(\mathrm{d}x) \to 0, \quad n \to \infty,$$

*if and only if*

$$\int |\phi_n(x)|\pi(\mathrm{d}x) \to \int |\phi(x)|\pi(\mathrm{d}x), \quad n \to \infty.$$

*Proof of Theorem 4.1.* Note that the Jacobian of the $\pi$-e.m.p $f$ is $\pi(x)/\pi(f^{-1}(x))$ by Proposition C.3, allowing the density of $\bar{q}_T$ to be expressed as:

$$\bar{q}_T(x) = \frac{1}{T} \sum_{t=1}^T f^t q_0(x) = \pi(x) \cdot \frac{1}{T} \sum_{t=1}^T \frac{q_0}{\pi}(f^{-t}(x)), \quad \forall x \in \mathcal{X}.$$

The pointwise density convergence is the direct consequence of Eq. (11). Specifically, provided $q_0 \ll \pi$, we have $q_0/\pi \in L^1(\pi)$, so the Birkhoff ergodic theorem [65; 64, p. 212] (see Theorem D.2) ensures:

$$\frac{1}{T} \sum_{t=1}^{T} \frac{q_0}{\pi} (f^{-t}(x)) \to 1, \qquad \pi - \text{a.e. } x \in \mathcal{X}, \qquad \text{as } T \to \infty. \tag{11}$$

The total variation convergence is then by the direct application of the *Scheffé's lemma* Lemma D.3. Notice that

$$\text{TV}(\widehat{q}_T, \pi) = \int \left| \frac{\widehat{q}_T}{\pi}(x) - 1 \right| \pi(\mathrm{d}x) = \int \left| \frac{1}{T} \sum_{t=1}^{T} \frac{q_0}{\pi} (f_\theta^{-t}(x)) - 1 \right| \pi(\mathrm{d}x).$$

To apply Lemma D.3, we set $\phi_t(x) := \frac{1}{T} \sum_{t=1}^{T} \frac{q_0}{\pi} (f_\theta^{-t}(x))$, and set $\phi(x) := 1$. Because $q_0 \ll \pi$, all $\phi_n$'s are $\pi$-integrable. Then, for all $n \in \mathbb{N}$, we obtain that

$$\int |\phi_n(x)| \pi(\mathrm{d}x) = \int \phi_n(x) \pi(\mathrm{d}x)$$

$$= \frac{1}{T} \sum_{t=1}^{T} \int \frac{q_0}{\pi} (f_\theta^{-t} x) \pi(\mathrm{d}x)$$

$$= \int q_0(\mathrm{d}x) \quad (\text{as } f_\theta \pi = \pi)$$

$$= 1 = \int |\phi(x)| \pi(\mathrm{d}x),$$

yielding the second convergence in Lemma D.3. $\qquad \square$

### D.3 Convergence of the IRF MixFlow

As hinted in the main text, the proof of Theorem 4.2 involves interpreting the IRF as a time-homogeneous, e.m.p. dynamical system on the joint space $\Theta^{\mathbb{N}} \times \mathcal{X}$. Specifically, we define a map $\Phi$ (Eq. (12)) whose iterates evolve both the state $X_t$ and the parameter sequence $(\theta_t)_{t \in \mathbb{N}}$. Overall, the proof proceeds in two steps. First, we show that the joint law of $(\theta_t, X_t)$ converges in total variation to $\mathbb{P} \otimes \pi$. Second, we deduce marginal convergence for $X_t$. Section D.3.1 establishes the joint result, while Section D.3.2 explains why it suffices to prove Theorem 4.2.

#### D.3.1 Convergence in the product space

The key technique for proving the joint convergence is to interpret the iterative process Eq. (2) as an autonomous, ergodic, and measure-preserving dynamical system in the joint space $\Theta^{\mathbb{N}} \times \mathcal{X}$. Given this framework, the joint convergence follows immediately, as substantiated by Xu et al. [39, Theorem 4.2] (which is based on the *mean ergodic theorem*).

For brevity, we define $\Omega = \Theta^{\mathbb{N}}$, $\mathcal{F}_{\mathbb{N}} = \mathcal{F}^{\otimes \mathbb{N}}$, and $\mathbb{P}$ be the joint distribution of $(\theta_t)_{t \in \mathbb{N}}$ with independent marginal distribution $\mu$. Define the *shift operator* $\sigma : \Omega \to \Omega$ by

$$\sigma \omega : (\omega_0, \omega_1, \dots) \mapsto (\omega_1, \omega_2, \dots).$$

And let $(\theta_n)_{n \in \mathbb{N}}$ be the coordinate process on $(\Omega, \mathcal{F}_{\mathbb{N}}, \mathbb{P})$, i.e., for all $\omega = (\omega_0, \omega_1, \dots) \in \Omega$,

$$\theta_n(\omega) = \omega_n.$$

By definition, we have $\theta_{n+1} = \theta_n \circ \sigma$, and $(f_{\theta_n})_{n \in \mathbb{N}}$ with $(\theta_n)_{n \in \mathbb{N}} \overset{\text{iid}}{\sim} \mu$ can be formally understood as $\left( f_{\theta_n(\omega)} \right)_{n \in \mathbb{N}}, \omega \sim \mathbb{P}$ satisfying that $f_{\theta_n(\omega)} = f_{\theta_0 \circ \sigma^n(\omega)} = f_{\theta_0(\sigma^n \omega)}$. For the rest of this work, we abuse the notation by writing $f_{\theta_n(\omega)}$ as $f_{\sigma^n \omega}$ for all $n \in \mathbb{N}$.

Now consider the product probability space $(\Omega \times \mathcal{X}, \mathcal{F}_{\mathbb{N}} \otimes \mathcal{B}, \mathbb{P} \times \pi)$, where $\mathbb{P} \times \pi$ denotes the joint distribution with independent marginals $\mathbb{P}$ and $\pi$ on $\Omega$ and $\mathcal{X}$ respectively. We define the transformation $\Phi : \Omega \times \mathcal{X} \to \Omega \times \mathcal{X}$ by

$$\Phi(w, x) = (\sigma \omega, f_{\sigma \omega}(x)), \quad \forall (\omega, x) \in \Omega \times \mathcal{X}. \tag{12}$$

Note that Eq. (12) equivalently describes the iterative process Eq. (2) with i.i.d. $(\theta_n)_{n \in \mathbb{N}}$. For the rest of the proof, we will focus on the autonomous dynamical system $(\Omega \times \mathcal{X}, \mathcal{F}_{\mathbb{N}} \otimes \mathcal{B}, \mathbb{P} \times \pi, \Phi)$.

**Theorem D.4.** *Under the same assumption of Theorem 4.2, we have*

$$\text{TV} \left( \frac{1}{N} \sum_{n=1}^{N} \Phi^n (\mathbb{P} \times q_0), \mathbb{P} \times \pi \right) \to 0, \quad \text{as } N \to \infty. \tag{13}$$

*Proof of Theorem D.4.* We first show that $\Phi$ preserves $\mathbb{P} \times \pi$, namely, $\Phi(\mathbb{P} \times \pi) = \mathbb{P} \times \pi$. For all $\xi \in L^1(\mathbb{P} \times \pi)$,

$$
\begin{aligned}
\Phi(\mathbb{P} \times \pi)(\xi) &:= \int_{\Omega \times \mathcal{X}} \xi(\omega, x) \Phi(\mathbb{P} \times \pi)(\mathrm{d}\omega, \mathrm{d}x) \\
&= \int_{\Omega \times \mathcal{X}} \xi \circ \Phi(\omega, x) \mathbb{P} \times \pi(\mathrm{d}\omega, \mathrm{d}x) \qquad (14) \\
&= \int_{\Omega} \int_{\mathcal{X}} \xi(\sigma\omega, f_{\sigma\omega}(x)) \pi(\mathrm{d}x) \mathbb{P}(\mathrm{d}\omega)
\end{aligned}
$$

Since $\sigma$ is measure-preserving for $\mathbb{P}$ due to the i.i.d. assumption, and $x \mapsto f_\omega(x)$ is $\pi$-measure-preserving by hypothesis, we obtain that

$$
\begin{aligned}
\Phi(\mathbb{P} \times \pi)(\xi) &= \int_{\Omega} \int_{\mathcal{X}} \xi(\omega, f_\omega(x)) \pi(\mathrm{d}x) \mathbb{P}(\mathrm{d}\omega) \\
&= \int_{\Omega} \int_{\mathcal{X}} \xi(\omega, x) (f_\omega \pi)(\mathrm{d}x) \mathbb{P}(\mathrm{d}\omega) \\
&= \int_{\Omega} \int_{\mathcal{X}} \xi(\omega, x) \pi(\mathrm{d}x) \mathbb{P}(\mathrm{d}\omega) \\
&= \int_{\Omega \times \mathcal{X}} \xi(\omega, x) \mathbb{P} \times \pi(\mathrm{d}\omega, \mathrm{d}x) \\
&=: (\mathbb{P} \times \pi)(\xi).
\end{aligned}
$$

This concludes that $(\Omega \times \mathcal{X}, \mathcal{F}_\mathbb{N} \otimes \mathcal{B}, \mathbb{P} \times \pi, \Phi)$ is a measure-preserving dynamical system.

We further show that $(\Omega \times \mathcal{X}, \mathcal{F}_\mathbb{N} \otimes \mathcal{B}, \mathbb{P} \times \pi, \Phi)$ is an ergodic dynamical system. Morita [46, Theorem 4.1] shows that it is equivalent to show the ergodicity of the shift dynamical system—$(\mathcal{X}^\mathbb{N}, \mathcal{B}^{\otimes \mathbb{N}}, \mathbb{P}_\pi, \tau)$—induced by the Markov process associated to Eq. (2). Here $\mathbb{P}_\pi$ is the unique probability measure on $(\mathcal{X}^\mathbb{N}, \mathcal{B}^{\otimes \mathbb{N}})$ so that the coordinate process $(X_1, X_2, \dots)$ is a Markov chain with kernel $P$ (Eq. (3)) and initial distribution $\pi$, and $\tau$ is the shift operator on $\mathcal{X}^\mathbb{N}$, i.e., $\tau(X_0, X_1, \dots) = (X_1, X_2, \dots)$. Douc et al. [66, Theorem 5.2.6] further guarantees that if $\pi$ is the unique invariant probability measure of $P$, then $(\mathcal{X}^\mathbb{N}, \mathcal{B}^{\otimes \mathbb{N}}, \mathbb{P}_\pi, \tau)$ is both measure-preserving and ergodic. Hence, the second assertion of Assumption 2.2 guarantees the ergodicity of $(\Omega \times \mathcal{X}, \mathcal{F}_\mathbb{N} \otimes \mathcal{B}, \mathbb{P} \times \pi, \Phi)$.

Finally, we apply Theorem 4.2 in Xu et al. [39] to finish the proof. Given that $\Phi$ is measure-preserving and ergodic for $\mathbb{P} \times \pi$, it remains to show that $q \ll \pi$ implies that $\mathbb{P} \times q \ll \mathbb{P} \times \pi$. For all $B \in \mathcal{B}$ and $F \in \mathcal{F}_\mathbb{N}$,

$$
0 = (\mathbb{P} \times \pi)(F, B) = \mathbb{P}(F) \times \pi(B) \implies \mathbb{P}(F) = 0 \text{ or } \pi(B) = 0.
$$

Since $(\mathbb{P} \times q)(F, B) = \mathbb{P}(F) \times q(B)$, if $\mathbb{P}(F) = 0$, then $\mathbb{P}(F) \times q(B) = 0$, and if $\pi(B) = 0$, then $q(B) = 0$ by hypothesis and $\mathbb{P}(F) \times q_0(B) = 0$ as well. Therefore, Xu et al. [39, Theorem 4.2] yields the desired result. $\qquad \square$

### D.3.2 From the joint convergence to Theorem 4.2

Finally, we justify why Eq. (13) is sufficient for Eq. (8).

*Proof of Theorem 4.2.* We first derive the explicit expression of $\Phi(\mathbb{P} \times q_0)$ and examine its conditional probability measure. Following the same derivation as Eq. (14), for all $\xi \in L^1(\mathbb{P} \times q_0)$,

$$
\begin{aligned}
\Phi(\mathbb{P} \times q_0)(\xi) &= \int_{\Omega} \int_{\mathcal{X}} \xi(\sigma\omega, f_{\sigma\omega}(x)) q_0(\mathrm{d}x) \mathbb{P}(\mathrm{d}\omega) \\
&= \int_{\Omega} \int_{\mathcal{X}} \xi(\omega, f_\omega(x)) q_0(\mathrm{d}x) \mathbb{P}(\mathrm{d}\omega) \\
&= \int_{\Omega} \int_{\mathcal{X}} \xi(\omega, x) (f_\omega q_0)(\mathrm{d}x) \mathbb{P}(\mathrm{d}\omega), \qquad (15)
\end{aligned}
$$

where the second equality is by the fact that $\sigma$ is measure-preserving for $\mathbb{P}$. Eq. (15) demonstrates that $\Phi(\mathbb{P} \times q_0)$ can be disintegrated into the marginal distribution $\mathbb{P}(\mathrm{d}\omega)$ on $\Omega$ and the conditional distribution $(f_\omega q_0)(\mathrm{d}x)$, yielding that

$$
X_n | (\theta_i)_{i \in \mathbb{N}} \sim f_{\theta_n} \circ \cdots \circ f_{\theta_1} q_0, \quad \text{for } n > 1,
$$

where $X_0 \sim q_0$. Hence, disintegration of $\frac{1}{N} \sum_{n=1}^{N} \Phi^n(\mathbb{P} \times q_0)$ on the slice $(\theta_1, \theta_2, \dots) \in \Omega$ is

$$
\frac{1}{N} \sum_{n=1}^{N} f_{\theta_n} \circ \cdots \circ f_{\theta_1} q_0.
$$

Then we show that the total variation convergence of the joint distribution (Theorem D.4) implies the total variation convergence of the conditionals (Theorem 4.2). For all $N \in \mathbb{N}$,

$$\mathrm{TV}\left(\frac{1}{N}\sum_{n=1}^{N}\Phi^n(\mathbb{P}\times q_0), \mathbb{P}\times\pi\right) = \int_{\Omega}\int_{\mathcal{X}}\left|\frac{1}{N}\sum_{n=1}^{N}\frac{\mathrm{d}\Phi^n(\mathbb{P}\times q_0)}{\mathrm{d}(\mathbb{P}\times\pi)} - 1\right|\pi(\mathrm{d}x)\mathbb{P}(\mathrm{d}\theta)$$

Notice that for all $n \in \mathbb{N}$, the Radon-Nikodym derivative $\frac{\mathrm{d}\Phi^n(\mathbb{P}\times q)}{\mathrm{d}(\mathbb{P}\times\pi)}$ always exists given that $\mathbb{P}\times q_0 \ll \mathbb{P}\times\pi$ and $\Phi$ is $\mathbb{P}\times\pi$-measure-preserving. And explicitly, since $\mathbb{P}\times q_0$ and $\mathbb{P}\times\pi$ have same marginal distributions on $\Omega$, we have

$$\frac{\mathrm{d}\Phi^n(\mathbb{P}\times q_0)}{\mathrm{d}(\mathbb{P}\times\pi)} = \frac{f_{\theta_n}\circ\cdots\circ f_{\theta_1}q_0}{\pi}.$$

Hence,

$$\mathrm{TV}\left(\frac{1}{N}\sum_{n=1}^{N}\Phi^n(\mathbb{P}\times q_0), \mathbb{P}\times\pi\right) = \int_{\Omega}\int_{\mathcal{X}}\left|\frac{1}{N}\sum_{n=1}^{N}\frac{f_{\theta_n}\circ\cdots\circ f_{\theta_1}q_0(x)}{\pi(x)} - 1\right|\pi(\mathrm{d}x)\mathbb{P}(\mathrm{d}\theta)$$

$$= \mathbb{E}\left[\mathrm{TV}\left(\frac{1}{N}\sum_{n=1}^{N}f_{\theta_n}\circ\cdots\circ f_{\theta_1}q_0, \pi\right)\right], \quad (\theta_n)_{n\in\mathbb{N}}\sim\mathbb{P}$$

Since $\mathrm{TV}(\cdot,\cdot)$ is always non-negative, the left-hand side converges to 0 as $N\to\infty$ yields that the following convergence holds in probability $\mathbb{P}$:

$$\mathrm{TV}\left(\frac{1}{N}\sum_{n=1}^{N}f_{\theta_n}\circ\cdots\circ f_{\theta_1}q_0, \pi\right) \to 0, \quad \text{as } N\to\infty.$$

This completes the proof. $\qquad\square$

## D.4 Convergence of the backward IRF MixFlow

*Proof of Theorem 4.3.* The pointwise density convergence is the direct consequence of Eq. (9) via Theorem 2.3. The total variation convergence is then established using identical strategy as the proof of Theorem 4.1 via Scheffé's lemma Lemma D.3. $\qquad\square$

## D.5 Convergence of the ensemble IRF MixFlow

*Proof of Theorem 4.4.* By the definition of the total variation,

$$\mathrm{TV}\left(\tilde{q}_T^{(M)}, \pi\right) = \int\left|\frac{\tilde{q}_T^{(M)}}{\pi}(x) - 1\right|\pi(\mathrm{d}x)$$

$$= \int\left|\frac{1}{M}\sum_{m=1}^{M}\frac{q_0}{\pi}\left(f_{\theta_1^{(m)}}^{-1}\circ\cdots\circ f_{\theta_T^{(m)}}^{-1}(x)\right) - 1\right|\pi(\mathrm{d}x).$$

By the triangle inequality,

$$\leq \int\left|\frac{1}{M}\sum_{m=1}^{M}\frac{q_0}{\pi}\left(f_{\theta_1^{(m)}}^{-1}\circ\cdots\circ f_{\theta_T^{(m)}}^{-1}(x)\right) - R^T\left(\frac{q_0}{\pi}\right)(x)\right|\pi(\mathrm{d}x) + \int\left|\int\frac{q_0}{\pi}(y)R^T\delta_x(\mathrm{d}y) - 1\right|\pi(\mathrm{d}x).$$

$$(16)$$

We derive upper bounds for two terms on the right-hand side separately.

For the first term, taking the expectation with respect to the randomness of $\theta\sim\mu$, and interchange the order of integrations,

$$\mathbb{E}\left[\int\left|\frac{1}{M}\sum_{m=1}^{M}\frac{q_0}{\pi}\left(f_{\theta_1^{(m)}}^{-1}\circ\cdots\circ f_{\theta_T^{(m)}}^{-1}(x)\right) - R^T\left(\frac{q_0}{\pi}\right)(x)\right|\pi(\mathrm{d}x)\right]$$

$$= \mathbb{E}\left[\int\left|\frac{1}{M}\sum_{m=1}^{M}\frac{q_0}{\pi}\left(f_{\theta_1^{(m)}}^{-1}\circ\cdots\circ f_{\theta_T^{(m)}}^{-1}(X)\right) - R^T\left(\frac{q_0}{\pi}\right)(X)\right|\mu\left(\mathrm{d}\theta_{1:T}^{(1:M)}\right)\right], \quad X\sim\pi$$

Notice that $\forall x\in\mathcal{X}$, $\left\{f_{\theta_1^{(m)}}^{-1}\circ\cdots\circ f_{\theta_T^{(m)}}^{-1}(x)\right\}_{m=1}^{M} \overset{\text{iid}}{\sim} R^T\delta_x$, where the randomness comes from the independent realization of $\theta$s, where $R$ is the induced the Markov process of $f_\theta^{-1}$. Therefore, applying Jensen's inequality yields

$$\leq \frac{1}{\sqrt{M}}\mathbb{E}\left[\sqrt{\mathrm{Var}_{\theta_{1:T}}\left[\frac{q_0}{\pi}\left(f_{\theta_1}^{-1}\circ\cdots\circ f_{\theta_T}^{-1}(X)\right)\mid X\right]}\right],$$

For the second term of Eq. (16), since $\frac{q_0}{\pi}$ is globally bounded by constant $B < \infty$, we have that

$$
\int \left| \int \frac{q_0}{\pi}(y) R^T \delta_x(\mathrm{d}y) - 1 \right| \pi(\mathrm{d}x)
$$

$$
= \int \left| \int \frac{q_0}{\pi}(y) R^T \delta_x(\mathrm{d}y) - \int \frac{q_0}{\pi}(y) \pi(\mathrm{d}y) \right| \pi(\mathrm{d}x)
$$

$$
\leq B \int \mathrm{TV}(R^T \delta_x, \pi) \pi(\mathrm{d}x)
$$

$$
= B \cdot \mathbb{E}\left[ \mathrm{TV}(R^T \delta_X, \pi) \right], \quad X \sim \pi.
$$

This completes the proof.

$\square$

## D.6 Proof of Proposition 3.1

*Proof of Proposition 3.1.* We first verify that the map defined in Algorithm 2 is $\bar{\pi}$-measure-preserving, invoking the second part of Proposition 3.1. The algorithm has four steps (see Section 3); we compute the Jacobian of each step. Steps 3-4 involve a discrete accept/reject decision, so we treat the two branches separately—within a branch the transformation is a diffeomorphism, making the Jacobian well defined.

1. Step 1 describes constant shifts applied to uniform random variables, which preserves $\mathrm{Unif}_{[0,1]}(\mathrm{d}u_v)$ and $\mathrm{Unif}_{[0,1]}(\mathrm{d}u_a)$ with Jacobian 1.

2. Step 2 is the CDF/inverse-CDF transformation of $\rho(\cdot|x)$. As long as the CDF $F(\cdot|x)$ is well-defined, this step describes a diffeomorphism in $\mathcal{V} \times [0,1]$. The corresponding Jacobian is given by:

$$
\frac{\rho(\widetilde{v}|x)}{\rho(v|x)}
$$

3. We analyze step 3 and 4 together. In the rejection branch, no additional transformation is applied, so the Jacobian is 1. In the acceptance branch, step 3 involves the involution mapping, with Jacobian $|\frac{\partial g(x,\widetilde{v})}{\partial x, \widetilde{v}}|^{-1}$, and step 4 rescale $u_a$ by the MH-ratio $r$, yielding a combined Jacobian with step 3 $\frac{\bar{\pi}(x',v')}{\bar{\pi}(x,\widetilde{v})}$.

Hence, in the rejection branch, the combined jacobian of step 1-4 evaluated on $s' = (x, \widetilde{v}, u'_v, u_a)$ is

$$
\frac{\rho(\widetilde{v}|x)}{\rho(v|x)} = \frac{\bar{\pi}(x, \widetilde{v}, u'_v, u_a)}{\bar{\pi}(x, v, u_v, u_a)}.
$$

In the acceptance branch, the combined jacobian of step 1-4 evaluated on $s' = (x', v', u'_v, u'_a)$ is

$$
\frac{\rho(\widetilde{v}|x)}{\rho(v|x)} \frac{\bar{\pi}(x', v')}{\bar{\pi}(x, \widetilde{v})} = \frac{\bar{\pi}(x', v', u'_v, u'_a)}{\bar{\pi}(x, v, u_v, u_a)}.
$$

Both satisfy the criterion of Proposition 3.1; the map is therefore $\bar{\pi}$-measure-preserving.

Finally, we show uniqueness of the invariant distribution. By Douc et al. [66, Corollary 9.2.16], an irreducible kernel has at most one invariant distribution. Because each $f_\theta$ preserves $\bar{\pi}$, the induced Markov kernel $P$ must admit $\bar{\pi}$ as an invariant distribution. If $P$ is irreducible, then $\bar{\pi}$ is its unique invariant distribution.

$\square$

# E  Additional experimental details

For all homogeneous MixFlows variants, the uniform-shift parameters were fixed to $\theta_v = \pi/8$ and $\theta_a = \pi/7$. For `NUTS` benchmarks, we use the Julia package `AdvancedHMC.jl` [84] with default settings throughout. The normalizing flow architectures were implemented as follows. In `RealNVP`, the affine coupling layers consist of two separate multilayer perceptrons (MLPs)—one for scaling and one for shifting—each with three fully connected layers and LeakyReLU activations. For `Neural Spline Flows (NSF)`, we set the spline bandwidth to $B = 30$, and used $K = 11$ knots. For synthetic examples, the hidden dimension in each MLP was set to 32 for `RealNVP` and 64 for `NSF`. For real-data examples, the hidden dimension was set to $\min(d, 64)$, where $d$ is the dimensionality of the target posterior distribution.

Experiments are conducted on the following platforms: a local machine equipped with an AMD Ryzen 9 5900X CPU and 64 GB of RAM, the ARC Sockeye computing platform at the University of British Columbia, and the high-performance compute cluster provided by the Digital Research Alliance of Canada. Code for reproducing the main experimental results is available at: `https://github.com/zuhengxu/MixFlow.jl.git`.

## E.1  Synthetic experiments

The four target distributions used in this experiment are as follows:

1. the banana distribution [76]:
$$y = \begin{bmatrix} y_1 \\ y_2 \end{bmatrix} \sim \mathcal{N}\left(0, \begin{bmatrix} 100 & 0 \\ 0 & 1 \end{bmatrix}\right), \quad x = \begin{bmatrix} y_1 \\ y_2 + by_1^2 - 100b \end{bmatrix}, \quad b = 0.1;$$

2. Neals' funnel [77]:
$$x_1 \sim \mathcal{N}\left(0, \sigma^2\right), \quad x_2 \mid x_1 \sim \mathcal{N}\left(0, \exp\left(\frac{x_1}{2}\right)\right), \quad \sigma^2 = 36;$$

3. a cross-shaped distribution: in particular, a Gaussian mixture of the form
$$x \sim \frac{1}{4}\mathcal{N}\left(\begin{bmatrix} 0 \\ 2 \end{bmatrix}, \begin{bmatrix} 0.15^2 & 0 \\ 0 & 1 \end{bmatrix}\right) + \frac{1}{4}\mathcal{N}\left(\begin{bmatrix} -2 \\ 0 \end{bmatrix}, \begin{bmatrix} 1 & 0 \\ 0 & 0.15^2 \end{bmatrix}\right)$$
$$+ \frac{1}{4}\mathcal{N}\left(\begin{bmatrix} 2 \\ 0 \end{bmatrix}, \begin{bmatrix} 1 & 0 \\ 0 & 0.15^2 \end{bmatrix}\right) + \frac{1}{4}\mathcal{N}\left(\begin{bmatrix} 0 \\ -2 \end{bmatrix}, \begin{bmatrix} 0.15^2 & 0 \\ 0 & 1 \end{bmatrix}\right);$$

4. and a warped Gaussian distribution
$$y = \begin{bmatrix} y_1 \\ y_2 \end{bmatrix} \sim \mathcal{N}\left(0, \begin{bmatrix} 1 & 0 \\ 0 & 0.12^2 \end{bmatrix}\right), \quad x = \begin{bmatrix} \|y\|_2 \cos\left(\text{atan2}\,(y_2, y_1) - \frac{1}{2}\|y\|_2\right) \\ \|y\|_2 \sin\left(\text{atan2}\,(y_2, y_1) - \frac{1}{2}\|y\|_2\right) \end{bmatrix},$$
where $\text{atan2}(y, x)$ is the angle, in radians, between the positive $x$ axis and the ray to the point $(x, y)$.

### E.1.1 Relative performance of homogeneous, IRF, and backward IRF MixFlows

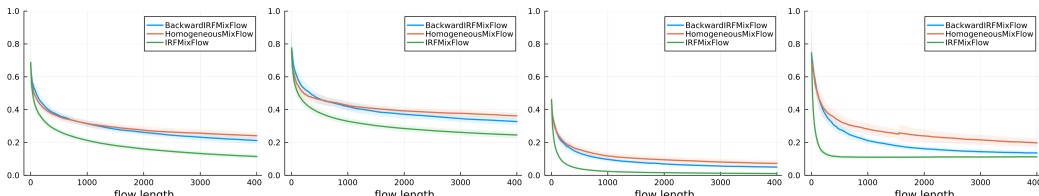

Total-variation error for homogeneous, IRF, and backward IRF MixFlows built on RWMH kernels, plotted against flow length T for the most performant step sizes among $\{0.05, 0.2, 1.0\}$. Each curve is the mean over 32 independent runs; shaded bands ($\pm 1$ SD) show run-to-run variability.

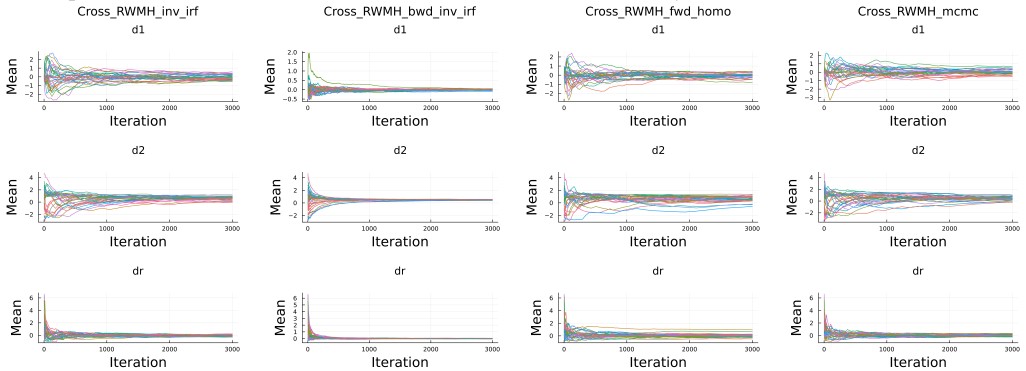

Running mean estimates over 3000 iterates from different IRF and MCMC dynamics based on RWMH, evaluated on the Cross distribution across 32 independent runs. Each line represents the trajectory of a single run. From top to bottom, the rows show the running mean of the test functions $(x_1, x_2) \mapsto x_1$, $(x_1, x_2) \mapsto x_2$, and $(x_1, x_2) \mapsto \frac{q_0}{\pi}(x_1, x_2)$. From left to right, the columns correspond to the dynamic of inverse IRF $f_\theta^{-1}$, the backward process of the inverse IRF, time-homogeneous dynamics $f_{\theta^\star}$, and the standard RWMH MCMC.

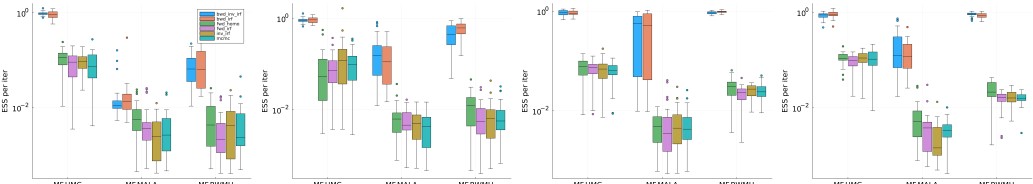

Per-sample MCMC effective sample size (ESS) estimates on the test function $\frac{q_0}{\pi}$, computed from trajectories generated by various IRF and MCMC dynamics based on HMC, MALA, and RWMH kernels. The trajectory lengths are set to 300 for HMC-based dynamics, 2000 for MALA, and 4000 for RWMH. Each ESS value is computed from a single trajectory, and the boxplots summarize the ESS estimates over 32 independent runs per method. The per-sample ESS for i.i.d. samples will be 1.

Figure 5: Results showing difference between homogeneous, IRF, and backward IRF MixFlows

Fig. 5a compares the total variation (TV) errors of homogeneous, IRF, and backward IRF MixFlows constructed from RWMH kernels. Overall, homogeneous and backward IRF MixFlows perform similarly, though the latter exhibits slightly improved accuracy at longer flow lengths. IRF MixFlow consistently outperforms both, achieving faster TV convergence and lower variability across runs. As discussed in Section 4.5, this improvement stems from differences in the convergence behavior of the series $\frac{1}{K} \sum_{k=1}^{K} \frac{q_0}{\pi} (T_k(x))$, where $T_k$ represents the sequence of transformations used in the density computation of each MixFlow variant.

Fig. 5b further illustrates this effect by showing running mean estimates over 3000 iterations for the Cross distribution. From top to bottom, each row shows the mean of the test functions $(x_1, x_2) \mapsto x_1$, $(x_1, x_2) \mapsto x_2$, and $(x_1, x_2) \mapsto \frac{q_0}{\pi}(x_1, x_2)$. From left to right, the columns correspond to the inverse IRF $f_\theta^{-1}$ (backward IRF MixFlow), the backward process of the inverse IRF (IRF MixFlow), the time-homogeneous flow $f_{\theta^\star}$ (homogeneous MixFlow), and standard RWMH MCMC. The backward process exhibits significantly faster convergence in all cases, consistent with the superior TV performance of IRF MixFlows under equal flow lengths. This advantage arises from reduced autocorrelation in the backward iterates.

Fig. 5c reports the per-sample MCMC effective sample size (ESS) for the test function $\frac{q_0}{\pi}$, estimated from trajectories generated using various IRF and MCMC dynamics based on HMC, MALA, and RWMH. This metric captures the degree of autocorrelation in $\frac{q_0}{\pi}(T_k(x))$ across iterations. Backward process dynamics consistently yield ESS values orders of magnitude higher than other methods—often approaching the ideal of independent sampling, with relative ESS close to 1 in some cases.

### E.1.2 Ensemble IRF MixFlows: scaling up $M$ or $T$

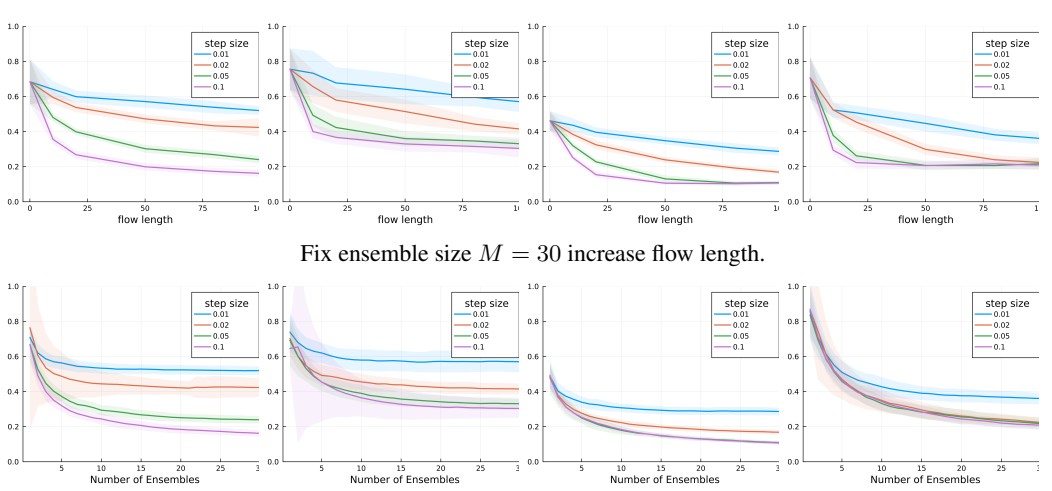

Fix ensemble size $M = 30$ increase flow length.

Fix flow length $T = 100$ increase ensemble size.

Figure 6: TV error of ensemble IRF MixFlows based on HMC over increasing ensemble size $M$ and flow length $T$. Each curve is the mean over 32 independent runs; shaded bands ($\pm 1$ SD) show run-to-run variability.

### E.1.3 Additional results for synthetic examples

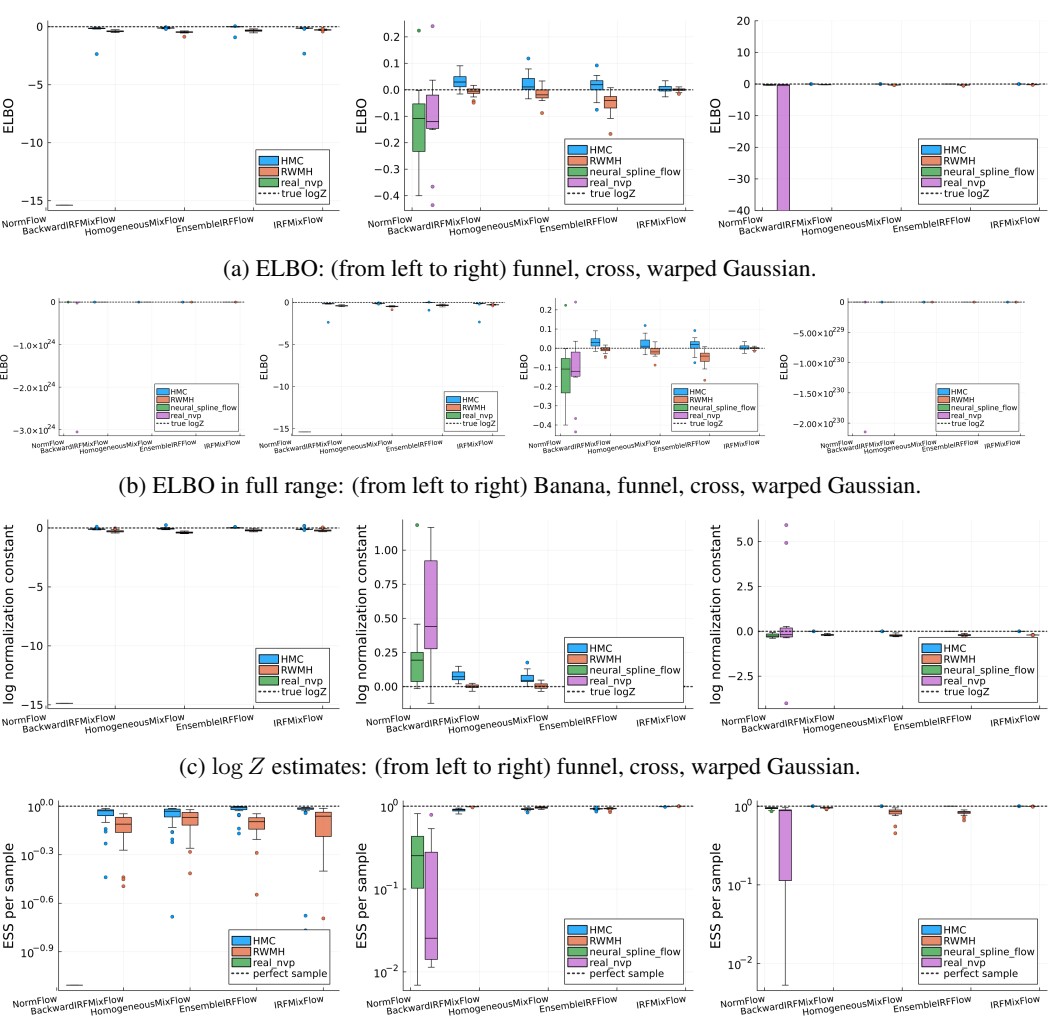

(a) ELBO: (from left to right) funnel, cross, warped Gaussian.

(b) ELBO in full range: (from left to right) Banana, funnel, cross, warped Gaussian.

(c) log $Z$ estimates: (from left to right) funnel, cross, warped Gaussian.

(d) Per-sample importance sampling ESS: (from left to right) funnel, cross, warped Gaussian.

Figure 7: Variational approximation quality of IRF Flows versus `RealNVP` and `NSF`. Box plots for IRF flows are based on 32 independent runs, and 10 runs for the normalizing flows.

## E.2   Additional results for real-data experiments

To approximate the ground truth, we ran an AIS procedure with 4096 particles with adaptive schedule selection. The initial temperature schedule was generated via mirror descent [85] with a small step size of 0.005; the schedule was then refined for five rounds using the adaptive scheme of Syed et al. [86], yielding more than 1000 annealing steps for each data set. All reference values are taken as the median estimates across 10 independent runs of the above procedure.

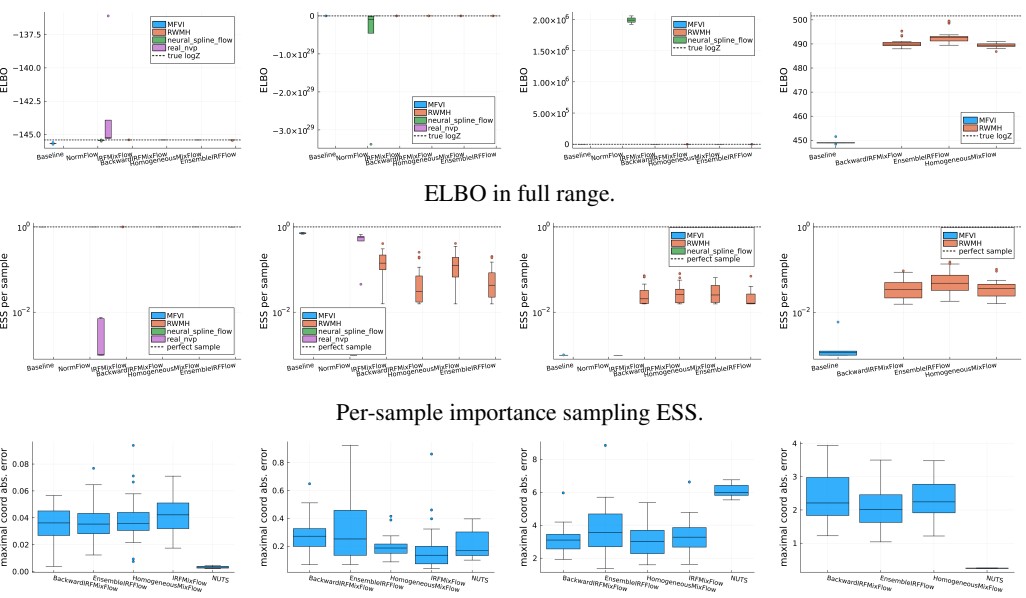

Figure 8: Results on real-data benchmarks (columns, from left to right): TReg($d = 4$), Brownian($d = 32$), SparseReg ($d = 83$), and LGCP ($d = 1600$)

