# OpenReview forum: "Asymptotically exact variational flows via involutive MCMC kernels"
_NeurIPS.cc/2025/Conference — NeurIPS 2025 poster_

### Official Review · Reviewer_zTkn · 2025-06-06

**Clarity:** 1
**Significance:** 2
**Originality:** 2
**Rating:** 4
**Confidence:** 4

**Summary:**

This paper aims at addressing convergence guarantees in variational inference (VI) models.
The quality of a variational approximation is fundamentally determined by the expressiveness of the variational model. Nonetheless, an expressive model is not necessarily guaranteed to converge due to the typical non-convexity of the optimization problem.
Conversely, MCMC methods are asymptotically exact but computationally intensive and not always differentiable or easily tunable. The paper tries to bridge this gap with this work. The authors introduce involutive MCMC kernels as invertible, measure-preserving iterated random functions (IRFs) acting as the flow maps of the (exact) variational flows.

The authors introduce some theorems to motivate their theorethical findings and validate their approach with extensive numerical experiments.

**Questions:**

- line 36: what do authors mean with "optimal tuning"?
- line 52-53: how about stochastic normalizing flows or recent methods combining MCMC sampling with training of variational families?
- eq(1) what is the difference between $x$ and $X$?
- I am having a hard time understanding the rationale behind eq (1): is this essentially representing an ensemble of flows, each of which comes with a different number of transformations/diffeomorphisms? How are these diffeomorphisms constructed in the first place?
- step 3 in Sec. 2.2: is the second term in the min a second order partial derivative or what does this notation represent?
- Sec 2.3: eq (2): Isn't this the same as a NF with multiple couplings? Could the authors clearly comment on differences and analogies between traditional normalizing flows and IRFs-based flows?
- can the author comment on what they mean by $\mu-a.s.$ and $\pi-a.e.$?
- In equation at bottom of page 3: this seems to be just the same as the average of normalizing Flow with increasing number of coupling layers, where $f_\theta_i$ represents a coupling layer with weights $\theta_i$. Can the authors comment on this intuition?
- Eq. 6 is reference on line 124 and later but never appears.
- I am confused by first equation in Sec. 3.2: Shouldn't $f_\theta$ act on $x$? How come it acts on the density $q_\theta$ in that equation? (Please reference all equations with numbers so that it is easier to identify them).
- First equation below line 151: I find the notation hard to parse and potentially not entirely consistent with the notation before. Shouldn't there be two sums in there? one running over t and one running over m?
- Lines 157/158: Do the authors there mean to average over different M flows all with fixed depth T? Does T have to be the same for every flow?
- Sec. 3.4 **relaxing ergodicity**: up to this point in the paper it is still very unclear to me where the MCMC involutive kernel contributes there. I believe it would be crucial for the authors to carefully revise the paper to make this concept emerging much clearer since the earlier sections of the manuscript.
- line 179: with using flow density and not $q_{\theta/T}$ here?
- The confusion stemming from section 3.4 still applies to me in Sec. 4. Can the authors try to re-elaborate in different words on the crucial differences between their approach and standard NFs method? I believe a related work section and also a more focused section on differences/analogies between their deterministic flows and NFs would be crucial to enhance clarity.
- Line 228-229: I am unsure if I get this statement. What makes an approach *reliably invertible*? To my understanding, only RWMH IRF seems to be reliably invertible, while all the other seems to quickly explode in errors after 100 steps.
- Line 243-244: Why not start from a standard normal distribution?
- Fig 4 is extremely hard to parse. Moreover, in some plots, it looks as if some approaches do not even appear in the plots. I wonder if this is a plotting issue or if the training was unsuccessful. If that is the case, I believe a more thorough discussion of these plots is needed. Perhaps it might be a good idea to carefully select a subset of these plots for the main text and thoroughly focus on those.

**Ethical Concerns:**

["NO or VERY MINOR ethics concerns only"]

**Final Justification:**

The authors' replies to my review and those from other reviewers have clarified many concern I had.
Nonetheless, I still believe the paper is hardly accessible to a broad audience and would greatly benefit from enhancing clarity in many parts of the manuscript (the authors themselves acknowledged that they will make parts of the manuscript more precise in the revised version). For this reason, I still believe the paper in its current form is not suited to the broad audience of NeurIPS.

On the other end, I was now able to fully appreciate and acknowledge the theoretical contribution of this work, hence the reason for raising my score.

To summarize, I will not fight to reject this paper should the other reviewers be strongly in favor for its acceptance, but at the same time I won't fight for it to be accepted either, in case the common agreement from the other reviewers is to reject the paper.

**Limitations:**

As far as I can tell, there's no section dedicated to the **Limitations** of the method, neither in the paper nor in the appendix.

**Paper Formatting Concerns:**

No major formatting issues detected.

**Quality:**

2

**Strengths And Weaknesses:**

# **Strengths**
- The paper addresses an important and timely topic in the field of variational inference.
- The experimental evaluation is comprehensive, covering a diverse range of benchmarks, including both synthetic and real-world datasets.

# **Weaknesses**
- The paper is difficult to follow at times and I believe is requires some efforts in enhancing clarity and accessibility.
- The notation is occasionally inconsistent, and some notations or acronyms are not properly defined.
- Certain concepts are assumed as known without sufficient explanation or background, which may hinder accessibility to a broader audience.
- The writing lacks clarity in presenting the main contribution. In particular, the significance of constructing *exact* variational families via involutive MCMC kernels does not come through clearly in the exposition.
- The visual presentation of experimental results is suboptimal. Figures are too small, as are the axis labels and legends. In addition, some plots lack sufficient detail in their captions (e.g., see Fig. 4).
- The conclusions drawn from the experimental results appear broader than the initial scope outlined in the abstract. This mismatch may confuse readers—for example, Fig. 2 is more confusing than illuminating, as it does not clearly support the central claim that the proposed methods outperform standard normalizing flows.
- Additional concerns are discussed further in the responses to the review questions.

---

> ### Author Rebuttal · Authors · 2025-07-30
>
> Thank you for reviewing our work! Please see our point-to-point response below.
> > **Weakness**:
> The paper is difficult to follow at times... //
> The notation is occasionally inconsistent... //
> Certain concepts are assumed as known without sufficient explanation or background...
>
> We would appreciate more specificity from the reviewer regarding the issues with writing,  notations and undefined acronyms. We are happy to revise the manuscript to improve the clarity.
>
> > ... the significance of constructing exact variational families via involutive MCMC kernels does not come through clearly in the exposition
>
> The main contributions of this work were stated as the 3 bullet points in the end of the introduction. And we point the review to the 3rd paragraph of the introduction on the significance of enabling the construction of MixFlows via general involutive MCMC kernels. As we explained explicitly in line 40-41, it is not clear how to design general invertible measure-preserving transformations for general continuous target distributions. This work addresses the limitation in full generality by constructing such transformations via involutive MCMC kernels.
>
> > The visual presentation of experimental results is suboptimal...
>
> Thank you for your feedback. We have improved the visualizations of the figures by adjusting its size and better curation of the reported results, and have included more informative captions for all figures in the manuscript.
>
> > The conclusions drawn from the experimental results appear broader than the initial scope outlined in the abstract. This mismatch may confuse readers—for example, Fig. 2 is more confusing than illuminating, as it does not clearly support the central claim that the proposed methods outperform standard normalizing flows.
>
> We respectfully disagree that our conclusions exceed the scope outlined in the abstract. First, Figure 2 does not involve any comparison with normalizing flows; as noted on lines 250–251, it simply contrasts the original Hamiltonian‑MixFlow—constructed with an uncorrected HMC kernel—with our corrected variant that includes the Metropolis–Hastings step. Second, our empirical results including comparisons against NFs (Figures 3 and 4) indeed support the abstract’s claim (lines 12–14) that our methods achieve competitive performance relative to standard normalizing flows.
>
> More crucially, it is not our central claim that our proposed methods outperform standard normalizing flows. The key contribution of this work, as stated in the abstract (line 3-4) is the general construction of an asymptotically exact variational family via involutive MCMC kernels.
>
> > **Questions**: what do authors mean with "optimal tuning"?
>
> This is an admittedly imprecise expression. We will revise this in camera-ready. We meant to state that the convergence of MixFlow does not rely on particular choice of its hyperparameters, reiterating its asymptotic exactness.
>
> > how about stochastic normalizing flows or recent methods combining MCMC sampling with training of variational families?
>
> This is a good question and we will include the discussion in camera-ready!
>
> These differentiabl AIS methods (DAIS; e.g., Wolf et al. 2016; Geffner & Domke 2021; Zhang et al. 2021; Thin et al. 2021b; Jankowiak & Phan 2021; Doucet et al. 2022; Geffner & Domke 2023) can be viewed as gradient-based tuning of AIS/SMC kernels or backward kernels. Unlike MixFlows, DAIS doesn’t produce an explicit density approximation, and its theory
> (Jankowiak & Phan 2021; Zhang et al. 2021; Zenn & Bamler 2024) typically operates under idealized assumptions—perfect transitions or optimal backward kernels—that do hold in practice or are only justifiable given an optimal tuning. These guarantees do not have the same generality of MixFlows’ asymptotic exactness.
>
> > eq(1) what is the difference between x and X?
>
> As defined by $ x \in \mathcal{X} $, x here denotes a point of the state space $ \mathcal{X} $. $ X \sim \bar q_T $ indicates that the random variable $X$ is distributed according to $ \bar q_T  $.
>
> > can the author comment on what they mean by $ \mu-a.s. $ and $ \pi-a.e. $ ?
>
> Here, “ \mu‑a.s.” stands for “\mu‑almost surely” and “\pi‑a.e.” for “\pi‑almost everywhere.” Both are standard shorthand in probability theory, meaning the stated property holds except on a set of measure zero (with respect to $ \mu $ or $ \pi $, respectively).
> >  ...the rationale behind eq (1):...
>
> Section 2.1 offers a concise primer on deterministic MixFlows.  Eq (1) formalizes the fact that a deterministic MixFlow is a mixture over flows obtained by applying a single ergodic measure‑preserving (e.m.p.) map $ f $ repeatedly to the base distribution $ q_0 $. In other words, each mixture component corresponds to $ f^t $ for some iteration count $ t $.
>
> For the concrete construction of the e.m.p. $ f $, as we explained in line 78–80, is not available in the literature for general continuous distribution and is indeed a major contribution of this work.
>
> > step 3 in Sec. 2.2: ...
>
> $ \left|\frac{\partial g(x, v)}{\partial(x, v)}\right| $ denotes the determinant of the Jacobian matrix of the involution map $ g $. We will denote this explicitly in camera-ready.
>
> > Sec 2.3: eq (2): Isn't this the same as a NF with multiple couplings? Could the authors clearly comment on differences and analogies between traditional normalizing flows and IRFs-based flows?
>
> We respectfully acknowledge that we’re unclear on what the reviewer means by “NFs with multiple couplings.” Equation (2) in Section 2.3 simply defines an IRF system, not a specific flow architecture.
>
> Broadly, the key differences between traditional normalizing flows (NFs) and our IRF‐based MixFlows are:
> (1). NFs apply a single invertible neural-network pushforward to a reference distribution; MixFlows form a mixture of multiple pushforwards.
> (2). NFs use target-agnostic flow maps, whereas MixFlows embed target information directly into each map, ensuring exact preservation.
>
> These design choices result in the crucial difference between these two families of flows: MixFlows are asymptotically exact, meaning that **every member within the family is guaranteed to converge to the target distribution as flow length increases**, independent of its hyperparameter tuning. Traditional NFs do not offer this convergence guarantee.
>
>
>
>
> > shouldn't $f_\theta$ act on $x$? How come it acts on the density...
>
> In this work, given a transformation $ f $ and a distribution $ p $, we write $ f p$ (f acts on the distribution $ p $) as the pushforward distribution of $ p $ through $ f $. We defined this notation in line 72.
>
> The equation $ \overrightarrow{q_T}:=\frac{1}{T} \sum_{t=0}^{T-1} f_{\theta_t} \circ \cdots \circ f_{\theta_1} q_0 $ denotes a mixture of $ T $ pushforward distributions $ \\{f_{\theta_t} \circ \cdots \circ f_{\theta_1} q_0: t = 0, \dots, T-1 \\} $. Same for the first displayed math of Section 3.2. The flow mapping $ f_{\theta_i} $ will be the invertible measure-preserving IRF modified from involutive MCMC kernels (as we detailed in section 4).
>
> > In equation at bottom of page 3: this seems to be just the same as the average of normalizing Flow with increasing number of coupling layers, where f_\theta_i represents a coupling layer with weights .
>
> $ f_{\theta_i} $ is not a coupling layer.
> As stated in line 101-103, $f_\theta$ is a set of parameterized bijection that preserves $\pi$. Thus, the parameterization of $f_\theta$ has to be informed by each particular $\pi$, and generic coupling flow layers wouldn’t suffice.
>
> > Eq. 6 is reference on line 124 and later but never appears.
>
> Eq (6) appears in the Apdx C.1 (line 827 of the PDF in the submitted Supplementary material). We will clarify this in the main text for explicitness.
>
> > First equation below line 151: I find the notation hard to parse and potentially not entirely consistent with the notation before. Shouldn't there be two sums in there? one running over t and one running over m?
>
> This is not an inconsistent notation nor a mistake. There is no sum running over $ t $. As mentioned in line 150-151, the ensemble IRF MixFlow is constructed by mixing over the final pushforwards of M IRF trajectories (sum over m), while all other IRF MixFlow variants mix along a single IRF trajectory (sum over t) .
>
> > Lines 157/158: Do the authors there mean to average over different M flows all with fixed depth T? Does T have to be the same for every flow?
>
> Yes, we use a fixed depth T for all M flows. Though one can use different length T for each flow if there are good reasons.
>
> > with using flow density and not $q_\theta/T$  here?
>
> This expression aims to explain shared patterns of the 4 variants of MixFlows. To avoid the collision of the notation, each MixFlow is denoted differently. And we use “flow density ” to indicate the density expression for all the four flows to avoid redundancy in words. We will edit this in camera-ready to improve the preciseness.
>
>
> > To my understanding, only RWMH IRF seems to be reliably invertible, while all the other seems to quickly explode in errors after 100 steps.
>
> We stated that HMC and MALA remain reliably invertible up to $ T \approx 200 $ iterations. Indeed, Figure 1 shows that by $ T = 200 $ the inversion error for both methods stays between $ 10^{-10} - 10^{-2} $. We believe that this aligns with our claim.
>
> > Why not start from a standard normal distribution?
>
> We adopted the same experimental setting from the original MixFlow work [26]. In general, it is recommended to choose a reference distribution $ q_0 $ to be close to the target distribution to avoid requiring excessive large flow length.
>
> > Fig 4 is extremely hard to parse...
>
> Yes, the excluded methods correspond to failure of the training. Again, we appreciate the reviews’ suggestion on improving the visualizations of the reported results. As responded above, we will work on this in revision.

---

### Official Review · Reviewer_UHcw · 2025-06-29

**Clarity:** 3
**Significance:** 3
**Originality:** 3
**Rating:** 5
**Confidence:** 4

**Summary:**

The authors introduce a IRF-based method for asymptotically exact variational inference that exempts the ergodicity assumptions from prior approaches. Towards this objective, they develop an involutive (self-invertible) MCMC kernel that parameterizes a measure-preserving IRF, which is shown to be invertible (Section 4) and asymptotically exact (Proposition 4.1). An ensemble algorithm with potentially better convergence rates is also proposed (Section 3.3).

In summary, the paper is well organized and nicely written, though somewhat dense. The contributions are clear and stated in a mathematically precise manner. However, the empirical analysis could be more comprehensive and rigorously executed.

**Questions:**

See weaknesses.

Also, traditionally, variational methods are characterized by finding the best tractable approximation to an intractable target distribution via gradient-based search. IRF-based approaches, however, take a different path. Could the authors discuss the connection between the proposed methodology and traditional VI?

**Ethical Concerns:**

["NO or VERY MINOR ethics concerns only"]

**Final Justification:**

I appreciate the authors' clear responses. I have increased my score to 5.

**Limitations:**

Limitations are properly addressed and future directions, e.g., designing automatically tuned and asymptotically exact variational methods, are concretely described.

**Paper Formatting Concerns:**

The paper is properly formatted.

**Quality:**

3

**Strengths And Weaknesses:**

## Strengths

1. The paper is theoretically sound and rigorously built upon the well-founded ergodic theory.

2. The work’s contributions and the limitations of its predecessors are clearly laid out and revisited throughout the paper.

3. Empirical results suggest that the proposed approach converges faster than traditional flow-based methods - while being asymptotically exact.

4. Computer code has been released for improved reproducibility.

5. Proofs for all statements and a simpler proof for the asymptotic exactness of MixFlows have been included in the appendix.

6. The proposed algorithm is, to the best of my knowledge, the first asymptotically exact variational method that doesn’t explicitly require ergodicity.

## Weaknesses

My concerns are mostly concentrated on the paper’s limited empirical analysis and on isolated claims. I will be happy to increase my score if the issues below are properly addressed.

1. Section 4 states that, “without loss of generality”, the IRF construction is described for a one-dimensional target distribution. It is unclear for me how such a construction can be straightforwardly generalized to higher dimensions. Could the authors elaborate on this extension?

2. Sections 5.1 and 5.2 report the results for the learning rate minimizing the TV distance and maximizing the ELBO, respectively. Although I acknowledge that this promotes a fair assessment of the proposed method, one is often interested in how the algorithm’s sensibility to hyperparametric changes. Authors should consider reporting the average and standard deviation of the selected metrics across runs.

3. Contrarily to NUTS, RealNVP, and NSF, the IRF variants seem to require a warm-up phase to appropriately define a step size for the RWMH kernel. It is unclear, however, whether this initial step is incorporated into the computational cost analysis in Figure 4(d). Also, I could not understand whether this hyperparameter is the same for every experiment or was independently chosen for each target distribution. Authors should clarify the cost of IRF tuning.

4. Synthetic experiments are based on 2-dimensional distributions that can be easily plotted. Could the authors show the resulting samples for each of the considered methods?

---

> ### Author Rebuttal · Authors · 2025-07-30
>
> Thank you for your insightful comments and for acknowledging our work! Please see our point-to-point response below.
>
> > Strength: ...
>
> Thank you again for your acknowledgement!
>
> > Questions:  Could the authors discuss the connection between the proposed methodology and traditional VI?
>
>
> We appreciate this important question, as it clarifies the positioning of our work. The MixFlow framework still lies in the traditional VI paradigm—i.e., approximating a target distribution within a chosen variational family. However, MixFlows uniquely define a variational family that is asymptotically exact, meaning **every member within the family is guaranteed to converge to the target distribution as flow length increases**, independent of its hyperparameter tuning. Therefore, unlike typical black-box variational families such as normalizing flows, the convergence of MixFlows does not critically depend on meticulous hyperparameter tuning—though different hyperparameter choices can still influence the approximation quality when using a finite number of flows.
> Moreover, since MixFlows are parameterized through MCMC kernels, their associated hyperparameters (e.g., step size) are typically low-dimensional and interpretable. This facilitates intuitive and effective tuning, such as targeting specific MH acceptance rates, which we have found to be consistently reliable across various examples. Nonetheless, developing more sophisticated and robust tuning procedures remains an interesting direction for future investigation.
>
> > Section 4 states that, “without loss of generality”, the IRF construction is described for a one-dimensional target distribution. It is unclear for me how such a construction can be straightforwardly generalized to higher dimensions. Could the authors elaborate on this extension?
>
> Thank you for highlighting this important point! We will clarify this extension and provide concrete examples (e.g., MALA, HMC, RWMH) for general dimensions in the final version of the paper (likely in the Appendix due to page limitations).
>
>
> To briefly illustrate, we elaborate here on how the four-step IRF construction (lines 194–198) naturally generalizes to higher dimensions, using RWMH as a concrete example. Importantly, steps 3 and 4 remain unchanged when moving to higher dimensions; only steps 1 and 2 require minor adjustments:
> - Step 1 (line 194): For a general d-dimensional target, both $u_v , \theta_v \in \mathbb{R}^d$, the step $ u_v \gets (u_v + \theta_v) mod 1$ is simply applied coordinate-wise.
> - Step 2 (line 195): In the specific case of RWMH (described in Example B.3, Appendix B.2), the distribution $ \rho(\mathrm{d} v |x) $ is a standard normal distribution $ N(0, I) $. Thus, we can implement step 2 in a coordinate-wise fashion as well. Specifically, for
> $$ i \in 1, \dots, d,  \quad [u_v’]i \gets F_i(v_i), \quad [\tilde v]i \gets F_i^{-1}([u_{v}]i) $$
> Where $ F_i $ and  $ F^{-1}_i $ represent, respectively, the CDF and inverse CDF of marginal (1-dimensional) standard normal distribution for the i-th coordinate.
>
> > Contrarily to NUTS, RealNVP, and NSF, the IRF variants seem to require a warm-up phase to appropriately define a step size for the RWMH kernel. It is unclear, however, whether this initial step is incorporated into the computational cost analysis in Figure 4(d). Also, I could not understand whether this hyperparameter is the same for every experiment or was independently chosen for each target distribution. Authors should clarify the cost of IRF tuning.
>
> Thank you for pointing this out! We've revised the manuscript to clarify this issue explicitly. Figure 4(d) indeed reports the **total computational cost**, including both the tuning phase (tuning step size for the RWMH kernel targeting at a 0.8 MH acceptance rate) and the subsequent inference. Additionally, we now have a figure comparing the total wall-clock time for all methods, showing a similar phenomenon as Figure 4(d). We will include the wall clock time comparison in the camera-ready.
> To further clarify, we did not reuse a single hyperparameter value across all runs. Instead, each independent run includes its own step-size tuning procedure.
>
> > Sections 5.1 and 5.2 report the results for the learning rate minimizing the TV distance and maximizing the ELBO, respectively. Although I acknowledge that this promotes a fair assessment of the proposed method, one is often interested in how the algorithm’s sensibility to hyperparametric changes. Authors should consider reporting the average and standard deviation of the selected metrics across runs.
>
> As mentioned in the previous response, in all real‐data experiments (Section 5.2) we evaluated every MixFlow run—32 independent trials—using the same tuning procedure, and we compared these to the normalizing‐flow (NF) results obtained under their best‐tuned settings (thereby giving NFs a practical advantage in our comparisons).
> For the synthetic benchmarks (Section 3.1), we plotted MixFlow performance across a range of kernel step sizes for the total variation metric (see Figure 2 in the main text and Figure 6 in Appendix C.1.2). Each curve represents the mean over 32 independent runs, and the shaded bands ($ \pm 1 $ SD) depict run‐to‐run variability. We believe these results sufficiently demonstrate MixFlow’s robustness to hyperparameter variations.
>
> > Synthetic experiments are based on 2-dimensional distributions that can be easily plotted. Could the authors show the resulting samples for each of the considered methods?
>
> Yes, we have obtained scatters of 1000 iid samples obtained from the IRF mixflows based on HMC kernels. The sample scatter matches the synthetic distribution almost perfectly. We will include this figure in the revised manuscript (perhaps included in the Apdx).
>
> As for a reference, the original deterministic MixFlow work (based on uncorrected HMC kernels) shows almost perfect sample scatters (see Figure 1 in [26]).

---

> > ### Comment · Reviewer_UHcw · 2025-08-03
> >
> > Thank you for the clear answers. I have increased my score to 5.

---

> > > ### Author Response · Authors · 2025-08-04
> > > **Thank you!**
> > >
> > > Thank you for raising the score and for taking the time to review our work. We appreciate your thoughtful feedback and support!

---

### Official Review · Reviewer_xH4x · 2025-07-01

**Clarity:** 2
**Significance:** 3
**Originality:** 3
**Rating:** 4
**Confidence:** 3

**Summary:**

The paper proposes a variational family for performing variational inference in Bayesian models, designed to have the theoretical convergence guarantees of MCMC. The key idea is to construct the variational family by generalising mix variational flows, which use Markov kernels to approximate the target distribution. The authors focus on involutive kernels and, to improve expressiveness and theoretical guarantees, the paper composes multiple involutive kernels using iterated random functions (IRFs), making random mix variational flows. These random mix flows relax the ergodic condition for convergence of the flow. Moreover, the method uses the backward process of the IRFs for training, arguing that this reduces computational cost. They further enhance their approach by proposing an ensemble of mix variational flows based on IRFs, and they provide theoretical results suggesting this ensemble can accelerate convergence. They introduce invertible versions of the kernel compositions and evaluate their performance on both synthetic and real datasets.

**Questions:**

Can you explain exactly how is it that the inverse of the backward IRF MixFlow has lower computational cost than the inverse of the forward IRF MixFlow?

How do the data benchmarks and computational cost of experiments compare to MALA or RWMH instead of NUTS? Maybe the computational cost figures would be clearer without the baseline. Maybe also present wall clock time?

What is the computing cluster used for experiments? is it a cluster of CPUs or GPUs? Using GPUs would parallelise many of the gradient and density evaluations of the computing cost. It would be interesting to know the wall clock time of training and inference of your method in comparison with normalizing flows when using parallel computing.

What are the computational costs of sample generation? especially comparing random mix flows with normalizing flows, or MCMC.

**Ethical Concerns:**

["NO or VERY MINOR ethics concerns only"]

**Final Justification:**

While I believe the experimental section could be strengthened, the core methodological contribution of the paper is both original and well executed. The proposed framework is meaningful in constructing asymptotically exact variational approximations, and for that reason, I believe the paper should be considered for acceptance.

**Limitations:**

There is no discussion on the limitations of the method. More about the computational costs, which are clearly expensive for an ensemble method but at what gain in expressiveness/convergence? and the limitations on architecture design and how this affects the outcome.

**Paper Formatting Concerns:**

No formatting issues.

**Quality:**

3

**Strengths And Weaknesses:**

Quality:
The paper is technically sound and builds on a foundation of existing theory, the proposal of using involutive kernels and their composition through IRFs is well motivated for the task of building variational families with weaker conditions for convergence. The practical algorithm presented for invertible measure-preserving IRFs makes the method applicable beyond theory. However, the work feels somewhat incomplete in places, particularly around the justification for using the backward IRF process, which is claimed to reduce computational cost but lacks convincing explanation or empirical evidence. The experiments do not fully explore the implications or variations of the proposed method, especially regarding architectural choices and computational costs. The authors acknowledge some limitations but could be more thorough in critically evaluating different design choices.

Clarity:
The paper is generally well organised and clearly written. However, the rationale and benefits of using the backward process of IRFs instead of the forward require better explanations. The limited discussion on implementation details, such as architecture choices and computational trade-offs for ensemble methods, reduces clarity for readers looking to apply the approach or reproduce results.

Significance:
By linking MCMC kernels, IRFs, and variational inference, the paper opens a promising direction for constructing flexible yet theoretically grounded approximations. The potential impact lies in enabling practitioners and researchers to build variational approximations that converge in total variation to the true posterior, addressing a known limitation of many variational approaches. The open questions regarding computational efficiency, backward vs. forward IRF usage, and ensemble trade-offs leave open how impactful the method will be in practice.

Originality:
While the paper largely combines and extends several existing ideas, it does so in a novel and meaningful way. The framing of variational families as compositions of involutive kernels via IRFs, particularly using the backward process, is an original contribution that advances understanding of how to construct variational approximations. The ensemble approach to accelerate convergence also adds novelty.

---

> ### Author Rebuttal · Authors · 2025-07-30
>
> Thank you for your review and insightful comments! Please see our point-to-point response below.
>
> > Summary: ... Moreover, the method uses the backward process of the IRFs for training, arguing that this reduces computational cost...
>
> We hope to clarify that we didn’t use any of the IRF for **training**. The IRF backward process is used to define the backward IRF MixFlow family. We remark that all variants of MixFlows are asymptotically exact regardless of tuning.
>
> > Question: Can you explain exactly how is it that the inverse of the backward IRF MixFlow has lower computational cost than the inverse of the forward IRF MixFlow?
>
> To make this precise, the density computation of the backward IRF MixFlow (Eq(5)) has lower computational cost than the density computation of the forward IRF MixFlow (Eq(9)).
> As explained in line 124-126, the latter requires simulating the backward process of the inverse IRF $ \overleftarrow{X_t}(x):=f_{\theta_1}^{-1} \circ \cdots \circ f_{\theta_t}^{-1}(x) \quad \text { for } t \in[T]$. Notice that we cannot compute this sequence sequentially—$ \overleftarrow{X_t}(x) \neq f_{\theta_t} \circ \overleftarrow{X_{t-1}}(x) $. Each $ \overleftarrow{X_t} $ has to be computed independently,  taking $O(T^2)$ operations to compute $ \\{\overleftarrow{X_t}(x): t\in [T]\\} $.
>
> In contrast, the density computation of the backward IRF MixFlow (eq(9)) requires simulating $  \\{f_{\theta_t}^{-1} \circ \cdots \circ f_{\theta_1}^{-1}(x): t \in [T] \\}$. Notice that the indexing is different from the aforementioned backward process. This process can be computed sequentially by applying $f_{\theta_t}$ in order, taking $ O(T) $ operations.
>
> We think this explanation suffices to justify the difference of the computational cost of the two IRF MixFlows, and are happy to make further clarification in camera-ready.
>
> > Quality: The experiments do not fully explore the implications or variations of the proposed method, especially regarding architectural choices and computational costs. The authors acknowledge some limitations but could be more thorough in critically evaluating different design choices.
>
> We appreciate your suggestion and fully agree that a rigorous understanding of the relative strengths and weaknesses among the four MixFlow variants is important. As noted in the main text (lines 177–183) and in Appendix E1.1.1–E1.1.2, we have already included empirical comparisons and discussions around the different MixFlow members and design choices. That said, we are happy to expand on these analyses in the camera-ready version if space permits.
>
> We would also like to clarify that a comprehensive investigation into the design choices and robust implementation strategies for MixFlows—such as more sophisticated tuning strategies for the MCMC kernel, and principled selection of M and T for the ensemble Mixflows—entails deeper methodological work. These are active research areas in Bayesian computation (e.g., [38-41, 47,48]) and are beyond the scope of the current submission. Our main goal in this work is to establish the foundational MixFlow framework and generic recipes of designing its flow maps so that we can have room for further refinement in future work.
>
> > Clarity: the rationale and benefits of using the backward process of IRFs instead of the forward require better explanations. The limited discussion on implementation details, such as architecture choices and computational trade-offs for ensemble methods, reduces clarity for readers looking to apply the approach or reproduce results.
>
> Please see our repsonse to the previous comment.
>
> > How do the data benchmarks and computational cost of experiments compare to MALA or RWMH instead of NUTS?
>
> HMC (NUTS) is known to outperform MALA in most settings (see, e.g., Chen and Gatmiry, 2023), so we do not expect MALA to offer more competitive performance on our benchmarks. Likewise, while RWMH may incur lower computational cost, it generally produces less accurate posterior estimates than NUTS.
> More importantly—as we note on lines 300–301—our goal is not to establish superiority over MCMC methods. We use NUTS as a benchmark to demonstrate that our asymptotically exact flows achieve comparable accuracy in estimating posterior summaries, while retaining the tractability of variational inference methods. We believe this choice of experiments is sufficient to support our claims.
>
> > Maybe the computational cost figures would be clearer without the baseline. Maybe also present wall clock time?
>
> This is a good suggestion! We now have a figure comparing the wall clock computation time of all the methods, which shows a similar phenomenon as the current Figure 4(d). For example, for the Brownian example, the wall clock time (in seconds) for each methods are shown as follows:
> | Deter/Bwd IRF MixFlow | Ensemble MixFlow    |NSF     |  RealNVP |  MFVI   |  NUTS   |
> | :-------- | :------- | :------ | :-------- | :------- |:------- |
> | ~ 2  |  ~14  | ~2600     |  ~ 784|    ~15 |  ~27   |
>
> In camera-ready, we will update Fig 4(d) with the wall clock time.
>
>
> > What is the computing cluster used for experiments? is it a cluster of CPUs or GPUs? Using GPUs would parallelise many of the gradient and density evaluations of the computing cost. It would be interesting to know the wall clock time of training and inference of your method in comparison with normalizing flows when using parallel computing.
>
> We conducted all experiments using CPU clusters, consistent with the wall-clock times we linked above. Detailed specifications of the computing clusters will be provided in the camera-ready version. Regarding the computational efficiency comparison between MixFlows and Normalizing Flows (NFs), we would like to clarify that our work does not claim superior computational efficiency for MixFlows. Indeed, NFs can significantly benefit from GPU-based parallelization. However, we emphasize that MixFlows—particularly ensemble MixFlows and IRF MixFlows as highlighted in our paper—can also leverage similar parallelizations.
>
> Crucially, our primary argument is centered around the fundamental difference between these two families of flows: MixFlows are asymptotically exact, whereas NFs are not. As a result, NFs typically require careful architecture selection and extensive hyperparameter tuning to avoid unreliable results or training failures. For instance, as demonstrated in Section 5.2 (Figure 4, third column), the training of RealNVP consistently diverged, returning NaNs across all configurations.  Similar challenges with NFs were also noted in the original MixFlow study (Section 6.3 of [26]) and in the RealNVP paper itself (Section 3.7 of [19]).
>
> In contrast, tuning of MixFlows is much easier as they are parameterized via MCMC kernels whose hyperparameters (such as step sizes) are more interpretable and mostly low-dimensional. We can simply tune these parameters based on MH acceptance rate (what we did in our experiments). For all our real data experiments, the step size tuning of RWMH based MixFlows is completed within 20 seconds to 1min, while NF training typically takes several hours to more than a day on the same hardware.
>
>
> > What are the computational costs of sample generation? especially comparing random mix flows with normalizing flows, or MCMC.
>
> If using the same involutive MCMC kernel, sample generation of the IRF MixFlows (simulating the forward process of IRF) is almost identical as running the MCMC iterations, but with the additional capabilities of weighing the samples based on the importance sampling weights with respect to the target.
>
> After training, the sample generation of NFs is typically much faster than MixFlows. But as we mentioned above, NFs come with significant initial tuning time and quality of samples relies on careful tuning. For example, in the Brownian example, by the time we have trained RealNVP, we have already run over 10million MCMC iterations (RWMH kernel).

---

> > ### Author Response · Authors · 2025-08-05
> >
> > We thank the reviewer once again for the valuable comments and suggestions. We hope that we have clearly addressed all questions, concerns, and weaknesses. Should you have any further questions, please do not hesitate to raise them.

---

> ### Comment · Reviewer_xH4x · 2025-08-07
>
> Thank you for your detailed and thoughtful responses, they’ve clarified several aspects and improved my understanding of the paper.
>
> Regarding the use of backward IRF MixFlows: based on your explanation, it seems this is the most practical variant. From my perspective, the paper would be easier to follow if the forward process were left to the appendix, with the main text focusing solely on the backward method. Just a suggestion.
>
> Also, while I understand your choice of NUTS, I’d like to point out that in high-dimensional models, the curse of dimensionality often affects NUTS more severely, and in my own experience, MALA tends to perform better. In fact, just from the plots in Figure 4, it seems that NUTS struggles in the sparse regression task, possibly due to divergence issues. So comparisons with MALA would be helpful.
>
> That said, I recognize your main contribution is methodological rather than computational. I’ll raise my score as I now believe the paper merits consideration for acceptance. I hope my comments help make the exposition even clearer in the final version.

---

> > ### Author Response · Authors · 2025-08-07
> > **Thank you for your continued response!**
> >
> > Thank you for your continued engagement and for your valuable comments! We appreciate your suggestions and the score increase.
> >
> > > From my perspective, the paper would be easier to follow if the forward process were left to the appendix, with the main text focusing solely on the backward method. Just a suggestion.
> >
> > Thank you for the suggestion. We’ve realized during the discussion period that our narrative and presentation could be clearer. As noted in our responses to Reviewers UFbd and zTkn, we plan to reorganize the manuscript for improved clarity.
> >
> > Regarding keeping only the backward IRF MixFlow in the main text, we need to give this more careful consideration. While the other three MixFlows do not suffer from the quadratic density‐computation cost of the forward IRF MixFlow, the parallelizable computation of the forward IRF MixFlow might address this issue. Further, given that the forward IRF MixFlow did show more desired approximation quality (see Apdx E.1.1), we think it's still valuable to be included in the maintext. We again appreciate your suggestions and will work to better curate the maintext material to improve clarity.
> >
> > >  I understand your choice of NUTS, I’d like to point out that in high-dimensional models, the curse of dimensionality often affects NUTS more severely, and in my own experience, MALA tends to perform better. In fact, just from the plots in Figure 4, it seems that NUTS struggles in the sparse regression task, possibly due to divergence issues. So comparisons with MALA would be helpful.
> >
> > Thank you for this explanation. It makes sense, and we may consider adding comparisons to MALA in the camera-ready version. However, the SparseRegression example is challenging not because of its dimensionality but due to the posterior’s varying geometry (e.g., funnel- or star-shaped). Such problems require samplers to adapt their step sizes locally to maintain an appropriate acceptance rate (see for example the discussion in Section 4.5 of [39]). For this reason, we suspect MALA won’t outperform NUTS.
> >
> > Finally, we’d like to emphasize that our goal is not to claim superiority over MCMC methods; the comparison against NUTS simply demonstrates that MixFlows can yield reasonable posterior summaries while also providing density approximations that are unavailable in general MCMC methods.

---

### Official Review · Reviewer_UFbd · 2025-07-02

**Clarity:** 3
**Significance:** 4
**Originality:** 4
**Rating:** 6
**Confidence:** 4

**Summary:**

The paper extends the recently introduced class of Mixflows algorithms - to construct variational approximations via ergodic measure preserving transformations - with non-deterministic maps. This allows for some results in terms of total variation convergence to the target distribution. They compare against normalizing flows and MCMC on estimating log normalization constants and certain posterior moments respectively.

**Questions:**

Additional comments/questions/suggestions for the authors:

- MixFlows and their extension here seem (to me, at least) clearly closely connected to adaptive importance sampling - which has a relatively well-developed literature - where exact density evaluations are provided and final estimators are often able to use all generated samples during the learning process. It would be good to comment on any connections that the authors see or do not see. Adaptive IS literature dates back to at least Pugh (1966), with other common references being Oh et al (1992), Bugallo et al (2017), Portier et al (2018). Of particular relevance here is that there exist adaptive IS algorithms where the proposal learning is driven by MCMC (Martino et al (2017)). Perhaps the authors could expand as it is sometimes possible to obtain central limit theorems for AIS estimates (see Portier et al (2018)), in contrast to estimates with MixFlows it seems.
- As it seems that all generated samples can be used in the final estimation here, it would be interesting to explicitly write down the Z (or log Z) estimate with importance sampling
- An empirical comparison with tempered SMC with HMC/NUTS kernels, seen by some as a very good reference for Z estimation, may strengthen further the relevance of the proposed approach ( default implementations exploiting GPUs are available in modern Python libraries, such as BlackJax), and commenting the advantages of the proposed approach (for example, not having to choose a sequence of distributions as in SMC). It may also be worth noting that SMC has been viewed also an adaptive importance sampling algorithm with intermediate target distributions.
- Aside of / instead of Theorem 2.2, I would have expected a theorem for the convergence of the normalizing constant estimates Z (or log Z thereof). After all $q_T$ for any $T$ does provide exact density evaluation, therefore any valid Z estimate would be naturally based on importance sampling, and this is what is done in the experiments as I understand. Further, the variational family would provide consistent estimates of expectations of integrable test functions even for finite $T$ when IS is used.
- In practice, is it easier to tune the MCMC kernels in your method than it is to tune the hyperaparameters of the normalizing flows training ? As in a real world scenario, that (difficult to quantify) cost may be also taken into account.
- Using the ELBO metric seems somewhat redundant (but perhaps the authors can explain) with estimation of log Z with importance sampling - as clearly the ELBO is intractable and has to be MC estimated anyway, so one may as well just refer to the estimation of log Z (which is of course a biased but consistent estimator the true log Z, with an MSE relating also to KL between target and proposal).
- Is Theorem 3.1. a convergence in distribution ? Please state it clearly.


References:
- Pugh EL. A gradient technique of adaptive Monte Carlo. SIAM Review. 1966 Jul;8(3):346-55.
- Oh, M.S. and Berger, J.O., 1992. Adaptive importance sampling in Monte Carlo integration. Journal of statistical computation and simulation, 41(3-4), pp.143-168.
- Bugallo, M.F., Elvira, V., Martino, L., Luengo, D., Miguez, J. and Djuric, P.M., 2017. Adaptive importance sampling: The past, the present, and the future. IEEE Signal Processing Magazine, 34(4), pp.60-79.
- Portier, F. and Delyon, B., 2018. Asymptotic optimality of adaptive importance sampling. Advances in neural information processing systems, 31.
- Martino, L., Elvira, V., Luengo, D. and Corander, J., 2017. Layered adaptive importance sampling. Statistics and Computing, 27, pp.599-623.

**Ethical Concerns:**

["NO or VERY MINOR ethics concerns only"]

**Final Justification:**

See my full comment below (which I wrote before realising I'd have to write here as well).

**Limitations:**

The authors have a fairly honest presentation and discuss limitations.

**Quality:**

4

**Strengths And Weaknesses:**

Strengths: the paper provides a nice continuation of the Mixflows research line with a nontrivial and interesting extension of this class of algorithms. The paper is largely clearly written, and the mathematical notation is generally precise. Experiments touch realistic posterior distributions and carefully take into account total computation cost (for instance, of training the neural networks in normalizing flows).

Weaknesses: Re-emphasizing this is a good paper, there can always be some reservations. Some parts remain partially unclear, although this may be a limitation of my understanding of iterated random function theory and/or Markov chain theory. In 3.2, is the simulation of the backward process required ? Reference [34] is cited, but it seems overkill to have to check it. Section 4 is also quite dense and hard to follow. Further, concrete examples of four MCMC-based IRFs arrive only in page 7. I recommend concrete instances to come earlier in the presentation of the IRF framework, and describe in a concrete way how they can be (approximately) inverted.
Further, the comment starting line 230: "Notably, the corrected HMC IRF is consistently more stable than its uncorrected counterpart used in past MixFlows work, because the additional MH accept reject step discards trajectories with large numerical error that would otherwise cause the dynamics to diverge." seems a bit directly at odds with the comment immediately after in line 235. Perhaps the authors can explain better here. Finally, the assumption in Theorem 3.3 of bounded (reverse) importance ratio with respect to the initial distribution seems strong, although I may be wrong. Could you comment in general about the importance of the choice of initial distribution ? (For instance, in SMC it matters quite a bit).

---

> ### Author Rebuttal · Authors · 2025-07-30
>
> Thank you for your insightful comments and for acknowledging our work! Please see our point-to-point response below.
>
> > **Weaknesses**: Re-emphasizing this is a good paper, there can always be some reservations. Some parts remain partially unclear, although this may be a limitation of my understanding of iterated random function theory and/or Markov chain theory.
>
> Thank you again for your acknowledgement and pointing out texts that deserve further clarification! We would revise the manuscript in camera-ready for a clearer explanation.
>
> > In 3.2, is the simulation of the backward process required ? Reference [34] is cited, but it seems overkill to have to check it.
>
> We will state this more clearly in revision. For the backward IRF MixFlow, you do not need to simulate the full backward process for either independent sample generation or density evaluation. Because $q_t$, drawing a sample simply involves selecting one mixture component uniformly at random, and applying that component’s forward map, avoiding simulating the whole backward process.
>
> > Section 4 is also quite dense and hard to follow. Further, concrete examples of four MCMC-based IRFs arrive only in page 7. I recommend concrete instances to come earlier in the presentation of the IRF framework, and describe in a concrete way how they can be (approximately) inverted.
>
> Thank you for the suggestion. In the revised manuscript, we will reorganize Section 4 to introduce MCMC‑based IRFs up front and include concrete examples—such as RWMH and HMC—to illustrate how these involutive MCMC algorithms admit an IRF representation and can be (approximately) inverted.
>
> > the comment starting line 230: "Notably, the corrected HMC IRF is consistently more stable than its uncorrected counterpart used in past MixFlows work, because the additional MH accept reject step discards trajectories with large numerical error that would otherwise cause the dynamics to diverge." seems a bit directly at odds with the comment immediately after in line 235.
>
> Thank you for highlighting this point! We will edit this paragraph in revision to resolve the confusion. To clarify, divergent chains are more unstable and exhibit drastically higher numerical inversion error. Although shadowing property does reduce the impact of such numerical error significantly, applying MH correction to filter out the unstable trajectories keeps the computation stable from the outset, so that all downstream inferences remain reliable without relying on shadowing.
>
> > the assumption in Theorem 3.3 of bounded (reverse) importance ratio with respect to the initial distribution seems strong, although I may be wrong. Could you comment in general about the importance of the choice of initial distribution ? (For instance, in SMC it matters quite a bit).
>
> This is a good question! We would like to clarify that the bounded density-ratio condition is not required to prove the asymptotic exactness of the ensemble MixFlow. For Theorem 3.3, however, we opted for a statement that intuitively highlights how the flow length $ T $ and the ensemble size $ M $ presents the bias-variance trade-off, rather than the tightest possible bound or the weakest assumption.
>
> Broadly speaking, the role of the initial distribution in MixFlow parallels its role in SMC or general MCMC: the closer $ q_0 $ is to $ \pi $, the fewer IRF iterations (T) are needed to reach a given approximation quality. Although our current theory establishes asymptotic exactness, it does not yet quantify the impact of $ q_0 $’s quality. We plan to conduct a more refined theoretical analysis that explicitly addresses the significance of the initial distribution. Thank you again for your insightful feedback!
>
>
> > **Questions**: MixFlows and their extension here seem (to me, at least) clearly closely connected to adaptive importance sampling...
>
> Thank you for highlighting the rich literature on adaptive multiple importance sampling—we weren’t previously aware of those specific references, and there are indeed many intriguing connections to our work.
>
> At a high level, both MixFlows and adaptive IS methods build a mixture approximation to the target distribution using proposals informed by MCMC samples. To our understanding, in adaptive IS (e.g., the Markov adaptive IS of Martino et al., 2017), one typically constructs a kernel‐smoothed density around a growing population of MCMC iterates, tuning kernel bandwidths via intermediate importance weights. By contrast, MixFlows directly parameterize a sequence of flow transformations using the Markov chain transitions themselves. This distinction leads to different theoretical guarantees: MixFlows enjoy pointwise density convergence and total‐variation convergence under minimal assumptions (thanks to their asymptotically exact nature), whereas adaptive IS approaches generally require additional conditions—such as an infinite particle limit and vanishing kernel scales—to attain similar convergence results.
>
> That said, adaptive IS does offer valuable ideas, particularly the use of intermediate weights and particles to adapt proposal hyperparameters. Incorporating a similarly robust adaptation mechanism for MixFlow kernel or step‐size parameters is an exciting direction for future work, and we will discuss this connection and potential extensions in the camera‐ready version.
>
> > As it seems that all generated samples can be used in the final estimation here, it would be interesting to explicitly write down the Z (or log Z) estimate with importance sampling
>
> Good point! We realize that we didn’t explicitly write out our Z estimates and we will make sure to clarify this in revision. For the current version, we just take iid samples from the MixFlow and do the following naive IS estimator based on iid draws:
> $$
> Z \approx \frac{1}{N}\sum_{n=1}^N  \frac{\pi}{q_T}(X_n), \quad (X_n)_{n=1}^N \stackrel{iid}{\sim} q_T
> $$
>
> However, we are aware that more sophisticated estimates can be obtained by taking advantage of the pushforward structure. For instance, for the deterministic MixFlow, we can use the trajectory averaged IS estimator (as noted in Algorithm 3 in [26]) that leverages all intermediate samples. We can also do the PISA estimator as investigated in [Andrieu et al, 2025] to use all intermediate samples. In this work, we try to restrict the scope to the construction of the variational family to avoid overcomplicating the narrative. A more comprehensive investigation of how to do estimates with the family is left for future work.
>
> > An empirical comparison with tempered SMC with HMC/NUTS kernels, seen by some as a very good reference for Z estimation, may strengthen further the relevance of the proposed approach...
>
> Thank you for the suggestion. We feel it may not fully clarify the relationship between our approach and SMC/annealed IS. On one hand, ensemble MixFlow does resemble annealed importance sampling—both propagate particles via parallel MCMC chains and assign weights, with performance driven by how well particles explore the target.  On the other hand, our method uses a homogeneous Markov kernel throughout, whereas SMC/annealed IS relies on inhomogeneous kernels guided by an annealing path.  This key distinction makes it difficult to disentangle the effect of the annealing path, and thus can make direct performance comparisons less informative. Therefore, understanding the connections to SMC/annealed IS requires a more careful investigation.
>
> That said, we are actively working on a follow‑up work that incorporates annealing and establishes a direct connection to annealed IS methods. Stay tuned!
>
> > Aside of / instead of Theorem 2.2, I would have expected a theorem for the convergence of the normalizing constant estimates Z (or log Z thereof). After all  for any  does provide exact density evaluation, therefore any valid Z estimate would be naturally based on importance sampling, ...
>
> Yes, you are totally correct that all standard guarantees of importance sampling would apply to MixFlows. As aforementioned, we are working on a followup work for a more comprehensive theory analysis and only keep theoretical results justifying the asymptotic exactness of MixFlows in this manuscript.
>
> We’d like to clarify that the purpose of Theorem 2.2 is to introduce the standard ergodic theorem of the IRF on any L1(pi) functions. This serves as the intuition on why we have pointwise density convergence of IRF MixFlows. For example, in eq(5) setting $ \phi $ (in the statement of Thm 2.2) to be $ \frac{q_0}{\pi} $ shows that $ \lim_{T \to \infty} \overleftarrow{q_T}(x)= \pi(x)$.
>
> > Using the ELBO metric seems somewhat redundant (but perhaps the authors can explain) with estimation of log Z with importance sampling...
>
> This is a fair point. We keep ELBO results in the main text mostly because it's the standard metric that’s used in the flow and VI community. Depending on the space, we might consider moving some ELBO results to Apdx in camera-ready.
>
> > Is Theorem 3.1. a convergence in distribution ? Please state it clearly.
>
> It is a  convergence in probability (with respect to the probability measure $ \mathbb{P} $). We will clarify this in camera-ready.

---

> > ### Author Response · Authors · 2025-08-05
> >
> > We thank the reviewer once again for the valuable comments and suggestions. We hope that we have clearly addressed all questions, concerns, and weaknesses. Should you have any further questions, please do not hesitate to raise them.

---

> ### Comment · Reviewer_UFbd · 2025-08-06
> **Response to authors**
>
> I thank the authors for their effort in addressing my comments.
> Thanks for the clarification about line 230.
>
> I would finally indeed recommend to write explicitly some Z estimates and include some discussion. Also, since you mention approaches that leverage intermediate samples (and also given the connection to adaptive IS), you may want to look into the relatively well-known "adaptive multiple importance sampling" of Cornuet et al (2012).
>
> Regarding your comment "adaptive IS approaches generally require additional conditions—such as an infinite particle limit and vanishing kernel scales—" - this would likely require a longer discussion (and not a very important point), but this comment seems very specific to some potentially only some (old?) adaptive IS approaches (KDE-based if I understand the comment well). Just as an example, the nonparametric method of Korba et al (2022) provides rates of convergence in KL divergence.
>
> Also, given the authors' already substantial work done (including mostly from other reviewers' requests) to improve the clarity of the paper, I increase my score (as I do believe the paper presents interesting ideas to be presented to the community).
>
> - Cornuet, J.M., Marin, J.M., Mira, A. and Robert, C.P., 2012. Adaptive multiple importance sampling. Scandinavian Journal of Statistics, 39(4), pp.798-812.
> - Korba, A. and Portier, F., 2022, May. Adaptive importance sampling meets mirror descent: a bias-variance tradeoff. In International Conference on Artificial Intelligence and Statistics (pp. 11503-11527). PMLR.

---

> > ### Author Response · Authors · 2025-08-07
> > **Thank you for your repsonse!**
> >
> > Thank you again for your valuable comments and continued engagement! We appreciate your recognition of our work and your score increase.
> >
> > > I would finally indeed recommend to write explicitly some Z estimates and include some discussion.
> >
> > Thank you for this suggestion. We will add an explicit discussion of our choices for  Z estimates---likely in the appendix for the camera-ready version due to page limitation.
> >
> >
> > > you may want to look into the relatively well-known "adaptive multiple importance sampling" of Cornuet et al (2012)/ ... nonparametric method of Korba et al (2022) provides rates of convergence in KL divergence.
> >
> > We’re grateful for these references. We were not aware of Korba et al. (2022) and find their annealing based methods very interesting!
> > Since the rebuttal, we started taking more comprehensive read on adaptive multiple IS (AMIS) literature. Since the rebuttal, we have begun a thorough review of the AMIS literature. As mentioned, many AMIS algorithmic ideas and theoretical analyses are applicable to developing robust adaptation routines for MixFlows. We will provide a discussion in the camera-ready!

---

### Decision · Program_Chairs · 2025-09-17

**Decision:**

Accept (poster)

**Comment:**

The authors give a general method for asymptotically exact variational inference using families of distributions obtained from normalizing flows, constructed from involutive MCMC kernels through representing them as iterated random function systems. They demonstrate improvement over previous methods.

Their method is general and flexible, resolving practical difficulties with previous methods.

Reviewers had a lot of questions, which the authors answered satisfactorily. All authors increased their ratings and recommend acceptance.
I remind the authors to incorporate the suggestions into the final manuscript, especially clarifications to be accessible to a broad audience.